# Evidence for loss and reacquisition of alcoholic fermentation in a fructophilic yeast lineage

Carla Gonçalves[1], Jennifer H Wisecaver[2,3], Jacek Kominek[4,5,6,7], Madalena Salema Oom[1,8], Maria José Leandro[9,10], Xing-Xing Shen[2], Dana A Opulente[4,5,6,7], Xiaofan Zhou[11,12], David Peris[4,5,6,7,13], Cletus P Kurtzman[14†], Chris Todd Hittinger[4,5,6,7], Antonis Rokas[2], Paula Gonçalves[1]*

[1]UCIBIO-REQUIMTE, Departamento de Ciências da Vida, Faculdade de Ciências e Tecnologia, Universidade Nova de Lisboa, Caparica, Portugal; [2]Department of Biological Sciences, Vanderbilt University, Nashville, United States; [3]Department of Biochemistry, Purdue Center for Plant Biology, Purdue University, West Lafayette, United States; [4]Laboratory of Genetics, University of Wisconsin-Madison, Madison, United States; [5]DOE Great Lakes Bioenergy Research Center, University of Wisconsin-Madison, Madison, United States; [6]J. F. Crow Institute for the Study of Evolution, University of Wisconsin-Madison, Madison, United States; [7]Wisconsin Energy Institute, University of Wisconsin-Madison, Madison, United States; [8]Centro de Investigação Interdisciplinar Egas Moniz, Instituto Universitário Egas Moniz, Caparica, Portugal; [9]Instituto de Tecnologia Química e Biológica António Xavier, Universidade Nova de Lisboa, Av. da República, Oeiras, Portugal; [10]LNEG – Laboratório Nacional de Energia e Geologia, Unidade de Bioenergia (UB), Lisboa, Portugal; [11]Integrative Microbiology Research Centre, South China Agricultural University, Guangzhou, China; [12]Guangdong Province Key Laboratory of Microbial Signals and Disease Control, South China Agricultural University, Guangzhou, China; [13]Department of Food Biotechnology, Institute of Agrochemistry and Food Technology (IATA), CSIC, Valencia, Spain; [14]Mycotoxin Prevention and Applied Microbiology Research Unit, National Center for Agricultural Utilization Research, Agricultural Research Service, U.S. Department of Agriculture, Peoria, United States

*For correspondence: pmz@fct.unl.pt

†Deceased

Competing interests: The authors declare that no competing interests exist.

**Abstract** Fructophily is a rare trait that consists of the preference for fructose over other carbon sources. Here, we show that in a yeast lineage (the *Wickerhamiella/Starmerella*, W/S clade) comprised of fructophilic species thriving in the high-sugar floral niche, the acquisition of fructophily is concurrent with a wider remodeling of central carbon metabolism. Coupling comparative genomics with biochemical and genetic approaches, we gathered ample evidence for the loss of alcoholic fermentation in an ancestor of the W/S clade and subsequent reinstatement through either horizontal acquisition of homologous bacterial genes or modification of a pre-existing yeast gene. An enzyme required for sucrose assimilation was also acquired from bacteria, suggesting that the genetic novelties identified in the W/S clade may be related to adaptation to the high-sugar environment. This work shows how even central carbon metabolism can be remodeled by a surge of HGT events.
DOI: https://doi.org/10.7554/eLife.33034.001

**eLife digest** Cells build their components, such as the molecular machinery that helps them obtain energy from their environment, by following the instructions contained in genes. This genetic information is usually transferred from parents to offspring. Over the course of several generations, genes can accumulate small changes and the molecules they code for can acquire new roles: yet, this process is normally slow. However, certain organisms can also obtain completely new genes by 'stealing' them from other species. For example, yeasts, such as the ones used to make bread and beer, can take genes from nearby bacteria. This 'horizontal gene transfer' helps organisms to rapidly gain new characteristics, which is particularly useful if the environment changes quickly.

One way that yeasts get the energy they need is by breaking down sugars through a process called alcoholic fermentation. To do this, most yeast species prefer to use a sugar called glucose, but a small group of 'fructophilic' species instead favors a type of sugar known as fructose. Scientists do not know exactly how fructophilic yeasts came to be, but there is some evidence horizontal gene transfers may have been involved in the process.

Now, Gonçalves et al. have compared the genetic material of fructophilic yeasts with that of other groups of yeasts . Comparing genetic material helps scientists identify similarities and differences between species, and gives clues about why specific genetic features first evolved.

The experiments show that, early in their history, fructophilic yeasts lost the genes that allowed them to do alcoholic fermentation, probably since they could obtain energy in a different way. However, at a later point in time, these yeasts had to adapt to survive in flower nectar, an environment rich in sugar. They then favored fructose as their source of energy, possibly because this sugar can compensate more effectively for the absence of alcoholic fermentation. Later, the yeasts acquired a gene from nearby bacteria, which allowed them to do alcoholic fermentation again: this improved their ability to use the other sugars present in flower nectars.

When obtaining energy, yeasts and other organisms produce substances that are relevant to industry. Studying natural processes of evolution can help scientists understand how organisms can change the way they get their energy and adapt to new challenges. In turn, this helps to engineer yeasts into 'cell factories' that produce valuable chemicals in environmentally friendly and cost-effective ways.

DOI: https://doi.org/10.7554/eLife.33034.002

## Introduction

Comparative genomics is a powerful tool for discovering links between phenotypes and genotypes within an evolutionary framework. While extraordinary progress in this respect has been observed in all domains of life, analyses of the rapidly increasing number of fungal genomes available has been particularly useful to highlight important aspects of eukaryotic genomes, including a broader scope of evolutionary mechanisms than was thus far deemed likely. For example, horizontal gene transfers (HGT) are thought to have played a very important role in domestication (*Gibbons et al., 2012*; *Marsit et al., 2015*; *Ropars et al., 2015*) and in the evolution of metabolism in fungi (*Alexander et al., 2016*; *Wisecaver and Rokas, 2015*). Instances of the latter are best showcased by the high frequency of HGT events involving gene clusters related to fungal primary and secondary metabolism (*Campbell et al., 2012*; *Khaldi and Wolfe, 2011*; *Slot and Rokas, 2010*; *2011*; *Wisecaver and Rokas, 2015*). When considering the horizontal transfer of single genes, those encoding nutrient transporters seem to be among the most frequently transferred (*Coelho et al., 2013*; *Gonçalves et al., 2016*; *Richards, 2011*). While the identification of HGT events can be straightforward given sufficient sampling of the lineages under study, inferences concerning the evolutionary driving forces behind HGT are often difficult and uncertain, because most HGT events identified are ancient. However, available evidence suggests that HGTs are often associated with rapid adaptation to new environments (*Cheeseman et al., 2014*; *Gojković et al., 2004*; *Qiu et al., 2013*; *Richards et al., 2011*; *Richards and Talbot, 2013*).

In line with these findings, we recently reported on the evolutionary history of a unique, high-capacity, specific fructose transporter, Ffz1, which is intimately associated with fructophilic metabolism in ascomycetous budding yeasts (subphylum Saccharomycotina) (*Gonçalves et al., 2016*).

Fructophily is a relatively rare trait that consists in the preference for fructose over other carbon sources, including glucose (*Cabral et al., 2015*; *Gonçalves et al., 2016*; *Sousa-Dias et al., 1996*). The evolution of *FFZ1* involved the likely horizontal acquisition of the gene from filamentous fungi (subphylum Pezizomycotina) by the most recent common ancestor (MRCA) of a lineage in the Saccharomycotina, composed so far entirely of fructophilic yeasts (*Gonçalves et al., 2016*). Most of the approximately one hundred species forming this clade (*Wickerhamiella* and *Starmerella* genera, as well as closely related *Candida* species), are associated with the floral niche and are often isolated from fructose-rich nectar (*Canto et al., 2017*; *de Vega et al., 2017*; *Lachance et al., 2001*). Interestingly, fructophilic lactic acid bacteria, whose metabolism has been dissected in detail, also populate the floral niche (*Endo et al., 2009*; *Endo and Salminen, 2013*). These bacteria have been shown to grow poorly on glucose, which can be at least partly explained by their lack of respiratory chain enzymes and alcohol dehydrogenase activity, deficiencies that hinder $NAD^+$ regeneration during growth on this sugar, as shown for *Lactobacillus kunkei* (*Maeno et al., 2016*). In contrast to glucose, fructose can be used both as a carbon source and as an electron acceptor for the re-oxidation of NAD(P)H (*Zaunmüller et al., 2006*), providing an explanation of why it is favored over glucose. Hence, fructophily in lactic acid bacteria seems to be linked to redox homeostasis (*Endo et al., 2014*). In yeasts, it is still unclear how preferential consumption of fructose may be beneficial, partly because unlike fructophilic bacteria, fructophilic yeasts grow vigorously on glucose when it is the only carbon and energy source available (*Sousa-Dias et al., 1996*; *Tofalo et al., 2012*). Our previous work suggested that, although a strict correlation was found so far between the presence of Ffz1 and fructophily in all species investigated (*Cabral et al., 2015*; *Gonçalves et al., 2016*; *Leandro et al., 2014*) and the requirement for *FFZ1* was genetically confirmed in the fructophilic species *Zygosaccharomyces rouxii* (*Leandro et al., 2014*), it is very likely that there are additional requirements for fructophily. Thus, the *FFZ1* gene does not seem to be sufficient to impart a fructophilic character to a previously glucophilic species.

To gain insight into the genetic underpinnings of fructophily in budding yeasts and how it may have become evolutionarily advantageous, here we used comparative genomics to identify traits, in addition to the presence of the *FFZ1* gene, that might differentiate yeasts in the fructophilic *Wickerhamiella/Starmerella* (W/S) clade, focusing on central carbon metabolism. Our results suggest that the evolution of fructophily may have been part of a process of adaptation to sugar-rich environments, which included a profound remodeling of alcoholic fermentation involving the acquisition of bacterial alcohol dehydrogenases, which turned out to be particularly important for glucose metabolism, and an invertase, which is essential for sucrose assimilation. In general, we found a surge of bacterial-derived HGT events in the W/S clade when compared with other lineages in the Saccharomycotina (*Marcet-Houben and Gabaldón, 2010*), many of which seem to impact redox homeostasis.

## Results

### The horizontally transferred Ffz1 transporter is essential for fructophily in *St. bombicola*

We previously reported the acquisition of a high-capacity fructose transporter (Ffz1) through HGT by the MRCA of W/S-clade species. This transporter was lost in the MRCA of the Saccharomycotina and was later transferred from a Pezizomycotina-related species to the MRCA of the W/S clade, and then from the W/S clade to the MRCA of the *Zygosaccharomyces* genus (*Gonçalves et al., 2016*). A putative role for Ffz1 in fructophily in the W/S clade was hypothesized based on its kinetic properties (*Pina et al., 2004*) and the evidence that it is indispensable for fructophily in the phylogenetically distant species *Z. rouxii* (*Leandro et al., 2014*). To test this hypothesis, a *FFZ1* deletion mutant was constructed in the genetically tractable W/S-clade species *Starmerella bombicola*. The sugar-consumption profile in YP medium supplemented with 10% (w/v) fructose and 10% (w/v) glucose (conditions where fructophily is apparent, hereafter referred to as 20FG medium), showed that fructophilic behavior was completely abolished in the *ffz1Δ* mutant (*Figure 1*), similarly to what was found in *Z. rouxii* (*Leandro et al., 2014*). A slight increase in the glucose consumption rate was also observed for the *ffz1Δ* mutant compared to the wild type when cultures were grown in 20FG medium (*Figure 1*).

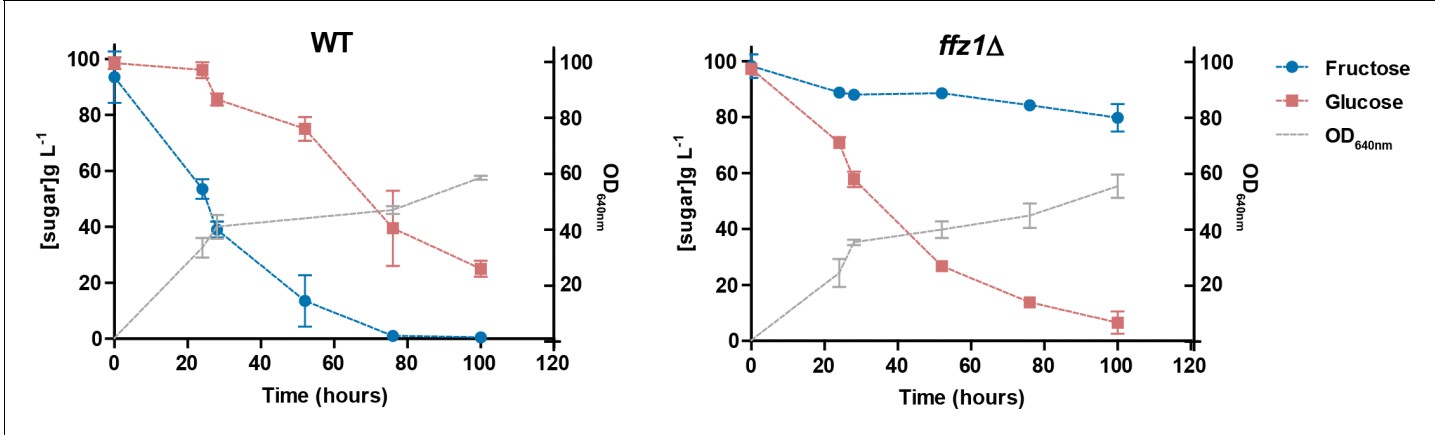

**Figure 1.** Sugar consumption profiles of *St. bombicola* wild type (wt) and *ffz1Δ*. Strains were grown in YP supplemented with 10% (w/v) fructose and 10% (w/v) glucose for 100 hr at 30°C with aeration. Error bars represent standard deviation of assays performed in duplicate in two biological replicates.
DOI: https://doi.org/10.7554/eLife.33034.003

The following source data is available for figure 1:

**Source data 1.** Data used to construct the plots is presented in *Figure 1*.
DOI: https://doi.org/10.7554/eLife.33034.004

## Alcoholic fermentation genes in W/S-clade yeasts

One of the most distinctive metabolic characteristics found in fructophilic bacteria that distinguished them from closely related non-fructophilic species was the lack of the enzymatic activity required for ethanol production and concomitant NAD(P)$^+$ regeneration. In one well-studied species, the gene encoding the bifunctional alcohol dehydrogenase (ADH)/aldehyde dehydrogenase (ALDH) normally responsible for ethanol production was absent (*Endo et al., 2014*), whereas in another species it was present but the encoded protein lacked the domain responsible for ethanol production (ADH), while maintaining the domain that conducts the preceding reaction (*Maeno et al., 2016*). On the other hand, some W/S-clade yeasts were previously known to be efficient producers of sugar alcohols or lipids to the detriment of ethanol (*Magyar and Tóth, 2011*;*Lee et al., 2003a*; *Kurtzman et al., 2010*). These observations prompted us to investigate whether alcohol dehydrogenase genes in fructophilic yeasts might also provide clues pertaining to a relation between fructophily and cofactor recycling in yeasts, as a first step toward unraveling other metabolic determinants of fructophily.

In *S. cerevisiae*, the *ADH1* gene encodes the enzyme mainly responsible for the conversion of acetaldehyde into ethanol (*de Smidt et al., 2008*). Hence, we started by retrieving homologs of *S. cerevisiae ADH1* from the genomes of six W/S-clade species as well as from four of their closest relatives (*Figure 2A*) using tBLASTx. Among the non-W/S-clade species considered, *Candida infanticola* (*Kurtzman, 2007*) occupies a particularly informative position, since it was phylogenetically placed as an outgroup of the W/S clade in our species phylogeny (*Figure 2A*), being its closest relative among the species included in this analysis. It has presently not been considered part of the W/S clade because it lacks the Ffz1 transporter (*Figure 2A*) and has not been isolated so far from sugar-rich habitats (*Kurtzman, 2007*). Notably, while the phylogenetic distance between all the species surveyed and *S. cerevisiae* was similar, protein sequence identity, *E*-value, and bitscore values denoted that predicted Adh1 proteins retrieved from W/S-clade species and *C. infanticola* as top hits of the tBLASTx search were much less similar to the *S. cerevisiae ADH1* query than the genes recovered from their non-fructophilic counterparts *Sugiyamaella lignohabitans*, *Blastobotrys adenini-vorans*, and *Yarrowia lipolytica* (*Figure 2B*). Moreover, when Adh1 protein sequences from W/S-clade species and *C. infanticola* were used as queries in BLASTp searches in the NCBI non-redundant (nr) database, the top 1000 hits consisted entirely of bacterial proteins, while when Adh1 sequences of *Su. lignohabitans*, *B. adeninivorans*, and *Y. lipolytica* yeasts were similarly employed as queries, the top 1000 hits recovered were fungal proteins. Taken together, these results suggest that W/S-

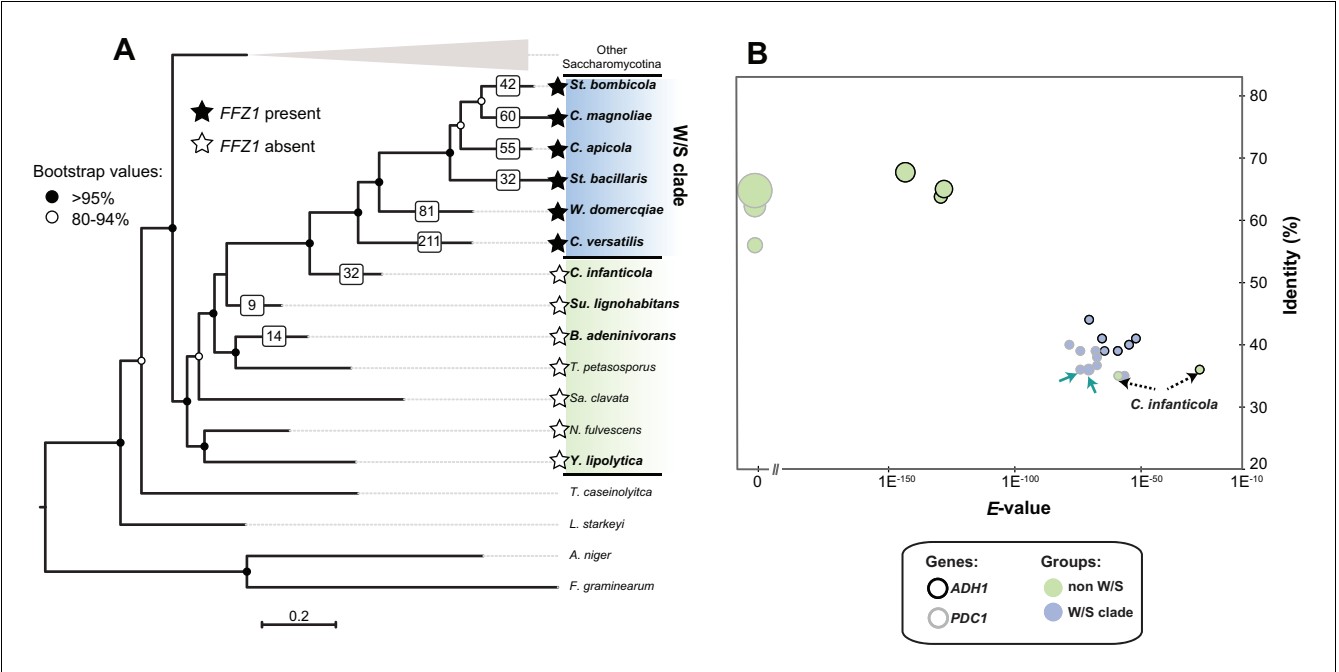

**Figure 2.** Maximum Likelihood phylogeny of Saccharomycotina (**A**) and tBLASTx results for alcoholic fermentation genes (**B**). (**A**) ML phylogeny depicting the phylogenetic relationship between W/S species (highlighted in blue) and closest relatives (highlighted in green); other Saccharomycotina lineages were collapsed as indicated. Names of the species are abbreviated as indicated in *Figure 2—source data 1*. Presence (black stars) and absence (white stars) of the Ffz1 transporter is indicated for each species. AI results are shown as the number of trees in which phylogenetic clustering with bacteria was confirmed for each species tested in the respective branch (white squares). (**B**) tBLASTx results for *ADH1* and *PDC1* searches in W/S-clade species and closest relatives *Su. lignohabitans*, *B. adeninivorans*, and *Y. lipolytica* (highlighted in bold in *Figure 2A*). For each gene and species, the plot depicts the *E*-value (x axis), identity (y axis), and bitscore (z axis, diameter of the circle) relative to the *S. cerevisiae* orthologs. *C. versatilis* Pdc proteins of bacterial origin are indicated by the blue arrows.

DOI: https://doi.org/10.7554/eLife.33034.005

The following source data and figure supplement are available for figure 2:

**Source data 1.** Accession numbers or coordinates for the proteins used to construct the species phylogeny.
DOI: https://doi.org/10.7554/eLife.33034.007

**Source data 2.** AI results for W/S species, *C. infanticola*, *B. adeninivorans*, and *Su. lignohabitans*.
DOI: https://doi.org/10.7554/eLife.33034.008

**Source data 3.** tBLASTx results for glycolytic proteins in the W/S clade.
DOI: https://doi.org/10.7554/eLife.33034.009

**Source data 4.** KEGG, Interpro and GO annotations of genes of bacterial origin in the W/S clade.
DOI: https://doi.org/10.7554/eLife.33034.010

**Figure supplement 1.** Complete ML phylogeny.
DOI: https://doi.org/10.7554/eLife.33034.006

clade species and *C. infanticola* have Adh1 homologs of bacterial origin, in contrast to the remaining three species.

The first step in the alcoholic fermentation pathway consists in the conversion of pyruvate in acetaldehyde, catalyzed by pyruvate decarboxylase (Pdc) (*Hohmann and Cederberg, 1990*). Since the gene encoding the 'native' enzyme catalyzing the second step, *ADH1*, is missing from W/S-clade genomes and seems to have been 'replaced' by a bacterial version, we next examined whether *PDC* genes mirrored somehow the peculiarities in the evolution of *ADH1* observed in the W/S clade. To this end, the sequence of the gene encoding the enzyme mainly responsible for conversion of pyruvate to acetaldehyde in *S. cerevisiae*, Pdc1, was used to retrieve its homologs in the set of species identified in *Figure 2A*. Remarkably, the Pdc sequences retrieved in this manner from the genomes of W/S-clade species and *C. infanticola* were also found to be more dissimilar to *S. cerevisiae* Pdc1 than those recovered from the three remaining non-fructophilic species, based on sequence identity, *E*-values, and bitscores (*Figure 2B*). In line with the observations for Adh1 sequences, two out of the

three Pdc sequences identified in *C. versatilis*, were more closely related to bacterial Pdc proteins than to fungal Pdc enzymes (*Figure 2B*). However, a BLASTp search using the third Pdc sequence from *C. versatilis* and the remaining Pdc sequences from W/S-clade species as queries, showed that their closest relatives were fungal proteins. In this case, the lower *E*-value appears to reflect the fact that the Pdc orthologs found in W/S-clade species and *C. infanticola* seem to belong to a decarboxylase family that is phylogenetically related to *S. cerevisiae* Aro10. In *S. cerevisiae*, Aro10 acts preferentially on substrates other than pyruvate and is not involved in alcoholic fermentation (*Kneen et al., 2011*; *Romagnoli et al., 2012*; *Vuralhan et al., 2005*). To better assess the phylogenetic relation between Aro10 and Pdc1-related sequences, and determine the evolutionary origin of the sequences identified in the W/S clade, a Maximum Likelihood (ML) phylogeny was reconstructed using the top 500 NCBI BLASTp hits using *S. cerevisiae* Pdc1 (CAA97573.1), *St. bombicola* putative Pdc ortholog, and *C. versatilis* Pdc sequences from apparent bacterial origin as queries. Putative Pdc sequences from the other W/S-clade species not available at the NCBI database were also included. This phylogeny (*Figure 3A and B*) confirmed the clustering of the W/S-clade sequences with Aro10 proteins from fungi, which indicates that *PDC1* was lost in the W/S clade. Additionally, as suggested by the BLASTp results, the two Pdc1-like proteins from *C. versatilis* were clustered with bacterial pyruvate decarboxylases (*Figure 3C*).

All glycolytic genes were examined in the same set of species and were all found to be present and to exhibit the expected level of similarity to *S. cerevisiae* query proteins (*Figure 2—source data 3*). This, together with the fact that other inspected publicly available genome assemblies of W/S-clade species are of very high quality (e.g. *Wickerhamiella domercqiae* JCM 9478, PRJDB3620 or *St. bombicola* JCM 9596 from RIKEN Center), makes it very unlikely that alcoholic fermentation genes were missed in W/S-clade species because of insufficient coverage or quality of the genome assemblies used.

## Identification of other genes of bacterial origin in the W/S clade

The tBLASTx analyses revealed that extant *ADH1* genes in W/S-clade species and also in *C. infanticola* were likely horizontally transferred from bacteria, and the same was observed for *PDC1* in *C. versatilis*. To identify other genes of bacterial origin in the W/S clade that might be related to central carbon metabolism and therefore to fructophily, a systematic high-throughput analytical pipeline based on the Alien Index (AI) score (*Alexander et al., 2016*; *Gladyshev et al., 2008*) was employed for HGT-detection in the previously defined group of species comprising W/S-clade representatives and close relatives outside the clade.

We found a considerably larger number of genes of bacterial origin in W/S-clade species and also in *C. infanticola* when compared with the two species more distantly related to the W/S clade (*Su. lignohabitans* and *B. adeninivorans*, *Figure 2A*, *Figure 2—source data 2*). Notably, *C. versatilis* displayed the highest number of putative HGT-derived genes from bacteria (211), followed by *W. domercqiae* with 80 genes for which phylogenetic clustering with bacteria could be confirmed (*Figure 2A*). Given that all W/S-clade species possess a higher number of HGT-derived genes when compared to their closest relatives, it is possible that a surge of HGT events occurred in the MRCA of the W/S clade. In line with this hypothesis, we predicted that it would be possible to identify a meaningful number of genes that were retained in more than one W/S-clade species. Indeed, after implementation of alignment thresholds (protein sequences > 150 amino acids) and collapsing the replicate phylogenies and lineage-specific gene duplications, it was possible to define 52 ortholog groups with representatives in two or more W/S-clade genomes. Not excluding the possibility that independent events also occurred in the various species, this suggests that a surge of HGT events took place in the MRCA of the W/S clade and that different species subsequently retained different sets of genes, *C. versatilis* being the species that retained the most genes of bacterial origin (*Figure 2A*). As expected, *ADH1* was present in this group (*Figure 2—source data 2*). *PDC1* was absent because only ortholog groups of bacterial origin detected in at least two W/S-clade species were selected, and bacterial Pdc1 proteins were only identified in *C. versatilis*. Another ortholog group relevant to fructose metabolism found among the 52 defined in this manner was *SUC2*, which encodes an invertase in *S. cerevisiae* that extracellularly hydrolyzes sucrose into fructose and glucose (*Carlson et al., 1981*). In *S. cerevisiae*, the *MAL* and the *IMA* genes have also been shown to play a role in sucrose hydrolysis (*Deng et al., 2014*; *Voordeckers et al., 2012*), but these genes are absent in the genomes of W/S-clade species, meaning that the horizontally transferred invertase appears to

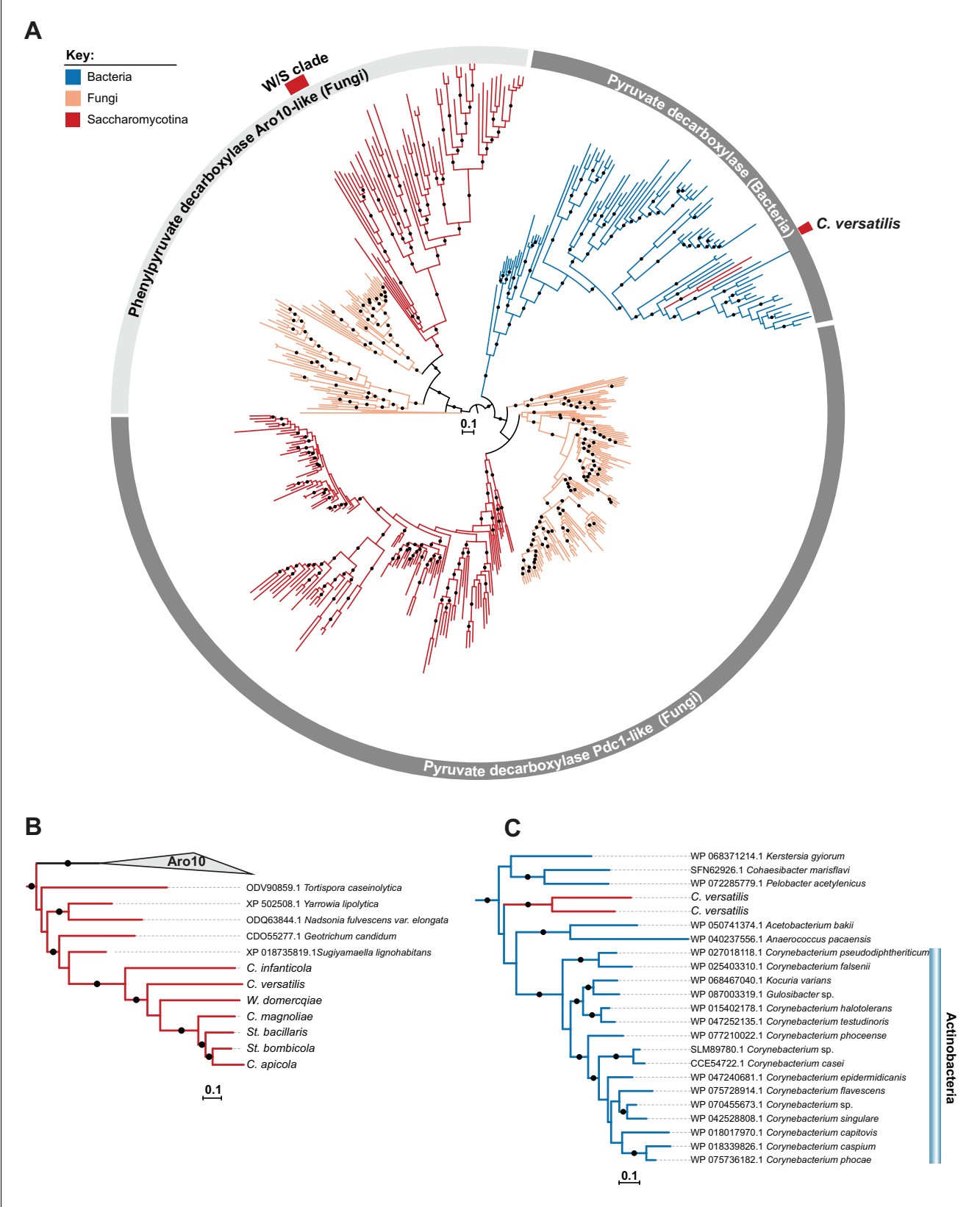

**Figure 3.** ML phylogeny of Pdc1 and Aro10. **A)** ML phylogeny of Pdc1-like proteins. W/S-clade and *C. versatilis* sequences are indicated by red blocks. Branches with bootstrap support higher than 90% are indicated by black dots. The different lineages are represented with different branch colors (red

*Figure 3 continued on next page*

*Figure 3 continued*
for Saccharomycotina, blue for bacteria and orange for other Fungi (i.e. non-Saccharomycotina). Clades highlighted in grey (Aro10- and Pdc1-like) were assigned according to the phylogenetic position of functionally characterized *S. cerevisiae* proteins. (B, C) Pruned ML phylogenies depicting the phylogenetic relationship between W/S-clade Aro10 proteins and their closest relatives in the Saccharomycotina (B) and between *C. versatilis* Pdc1 xenologs and the closest related bacterial pyruvate decarboxylases (C). For W/S- clade sequences, protein ID is indicated before the abbreviated species name.
DOI: https://doi.org/10.7554/eLife.33034.011

be the only enzyme with sucrose-hydrolyzing capacity encoded in the W/S-clade genomes investigated. In a phylogenetic tree reconstructed using the 200 top phmmer hits to W/S-clade species *Candida magnoliae* (protein ID 2301, see *Figure 2—source data 2*), strong support for clustering of W/S-clade sequences to Acetobacteraceae SacC sequences was observed (*Figure 4A* and *Figure 4B*). A topology comparison test (Approximately Unbiased, AU) also strongly supported the HGT hypothesis (p-value=$4e^{-07}$).

To check for putative enrichment in other protein functions among the remainder of the transferred genes, the 52 ortholog groups were subsequently cross-referenced with the Kyoto

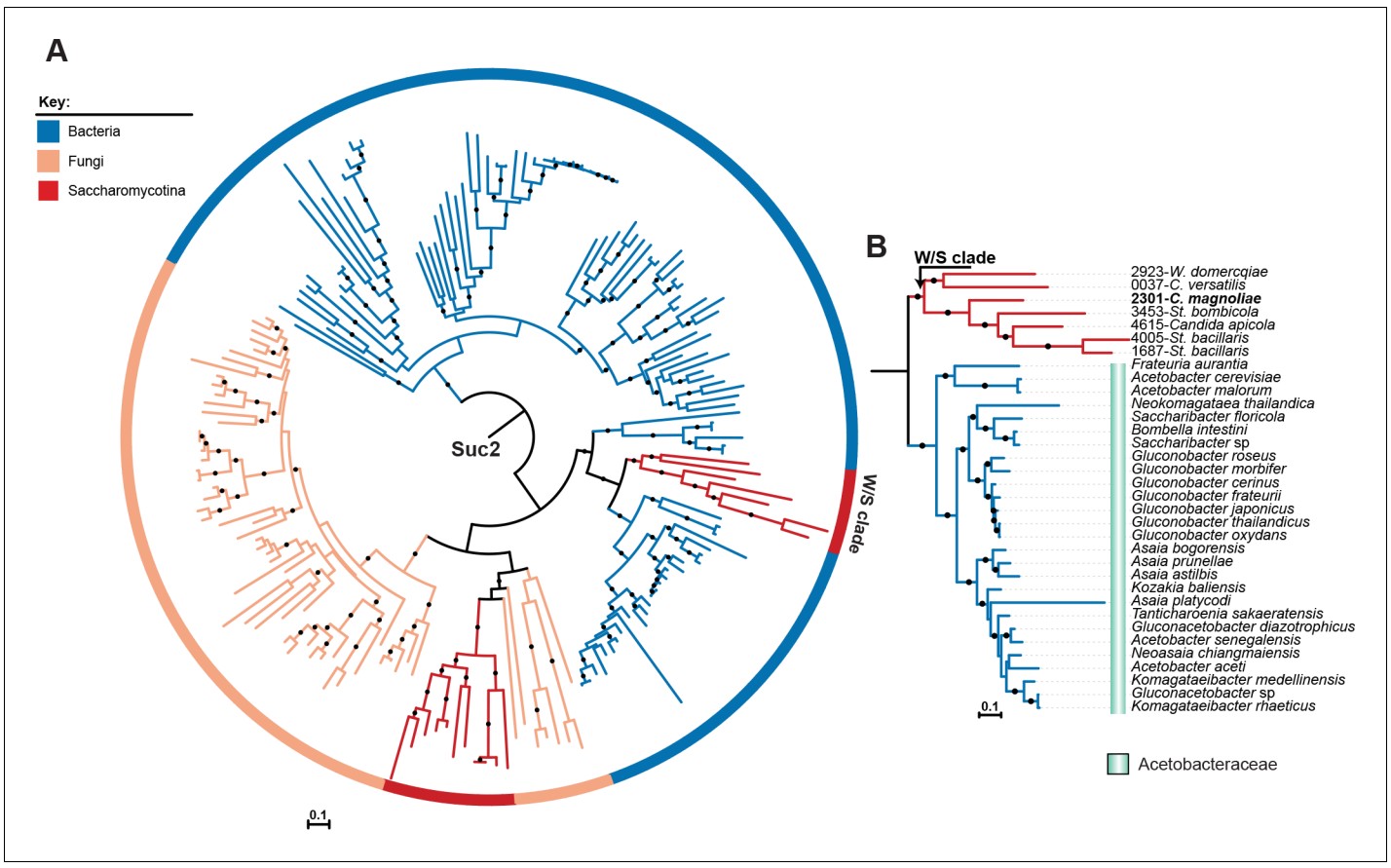

**Figure 4.** ML phylogeny of fungal Suc2 and bacterial SacC proteins. A) ML phylogeny of Suc2/SacC proteins (top 200 phmmer hits). W/S-clade species are highlighted. The different lineages are represented with different branch colors (red for Saccharomycotina, blue for bacteria, and orange for other Fungi (i.e. non-Saccharomycotina)). Branches with bootstrap support higher than 90% are indicated by black dots. (B) Pruned ML phylogeny of Suc2 depicting the phylogenetic relationship between the W/S clade and Acetobacteraceae. For W/S sequences, protein ID is indicated before the abbreviated species name.
DOI: https://doi.org/10.7554/eLife.33034.012

The following figure supplement is available for figure 4:

**Figure supplement 1.** Growth and consumption of sucrose in *St*.
DOI: https://doi.org/10.7554/eLife.33034.013

Encyclopedia of Genes and Genomes (KEGG), Gene Ontology (GO), and InterPro annotations. Notably, out of 43 proteins to which a GO molecular function was assigned, 16 impacted redox homeostasis and were associated with oxidoreductase activity (GO:0016491 and GO:0016616, 14 genes) and peroxidase activity (GO:004601, two genes), while a BLAST KOALA annotation (*Kanehisa et al., 2016*) indicated that the biological processes most frequently involved were amino acid metabolism, carbohydrate metabolism, and metabolism of cofactors and vitamins (*Figure 2—source data 4*). Notably, some of these genes appeared to have undergone several intraspecific duplications, in particular those encoding oxidorreductases participating in various metabolic pathways.

## Evolution of alcohol dehydrogenase genes in W/S-clade species

We noted that, in addition to *ADH1*, other alcohol dehydrogenase genes seem to have been horizontally acquired by W/S-clade species (*Figure 2—source data 4*), including putative *ADH6* and *SFA1* orthologs, which can also participate in alcoholic fermentation in *S. cerevisiae* (*Drewke et al., 1990*; *Ida et al., 2012*). In all cases, except for *SFA1,* the 'native' yeast orthologs appear to have been lost in W/S-clade genomes. This could imply that ethanol production is conducted by alcohol dehydrogenases of bacterial origin in W/S-clade species. To learn more about the evolutionary history of these genes, detailed phylogenetic analyses were conducted for Adh1 and Adh6, for which maximum likelihood phylogenies were reconstructed using the top phmmer hits obtained using *St. bombicola* Adh1 and Adh6 proteins as queries.

The resulting Adh1 tree (*Figure 5A*) included protein sequences from both bacteria and fungi. All W/S-clade species clustered with strong support with the Acetobacteraceae (Proteobacteria) Adh1 proteins (*Figure 5A* and *Figure 5B*). Within the Adh1 W/S-clade cluster, the overall phylogenetic relationships were in line with the expected relationships between the species (*Figure 2A*), suggesting that a single HGT event occurred in the MRCA of this clade. Topology comparison tests (AU) strongly supported the Adh1 HGT event to the W/S clade (p-value=8e$^{-03}$), adding to the strong evidence of HGT provided by the robustly supported branch that clusters the W/S-clade xenologs with bacteria and the AI results. Adh1 sequences from *C. infanticola* also clustered with those of proteins of bacterial origin. However, the two Adh1 sequences from *C. infanticola* are not similar to those of the W/S clade (*Figure 5C*), as might be expected if a single HGT event were responsible for the acquisition of the *ADH1* gene in both lineages. These sequences instead grouped, albeit with weak support, with Adh1 sequences from the distantly related Lactobacillales and Enterobacteriales (*Figure 5C*), implying that an independent HGT event may have occurred in the *C. infanticola* lineage. Nonetheless, topology tests did not strongly support an independent origin for W/S and *C. infanticola* Adh1 proteins (p-value=0.073).

An extended Adh6 ML phylogeny (*Figure 6A*) was reconstructed with the top 10,000 phmmer hits to show that the W/S-clade sequences are indeed Adh6 orthologs. In this phylogeny, W/S-clade sequences clustered with strong support (>95%) with Proteobacteria Adh6 sequences, within a large cluster that also encompasses known fungal Adh6 proteins. Remarkably, while Adh1 W/S-clade sequences grouped with those of the Acetobacteraceae (*Figure 5A* and *Figure 5B*), the Adh6 sequences are more closely related to those of other bacterial families, as highlighted in *Figure 6B*. The phmmer search failed to uncover Adh6 sequences in the *C. infanticola* proteome. The absence of an *ADH6* ortholog was further confirmed by a tBLASTx search against the *C. infanticola* genome using Adh6 sequences from both *S. cerevisiae* (KZV09178.1) and *St. bombicola* as queries. This result suggests that both *ADH1* and *ADH6* were lost in an ancestor of *C. infanticola* and the W/S clade. Interestingly, the *ADH6* xenologs were apparently duplicated several times within each W/S-clade species (*Figure 6A*, *Figure 6B* and *Figure 2—source data 4*), and *Starmerella bacillaris* and *C. magnoliae* harbor the most paralogs (four in total).

## Kinetic properties of alcohol dehydrogenases of bacterial origin in W/S-clade species

Acquisition of bacterial alcohol dehydrogenases by W/S-clade yeasts could have been driven by putative benefits afforded by an enzyme with kinetic characteristics that provide some advantage that the 'native' enzyme presumably lacked or, alternatively, by the need to restore alcoholic fermentation after an ancestral loss event, possibly in connection to adaptation to a new environment. To help elucidate this and also taking into account the link found between Adh activity and fructophily

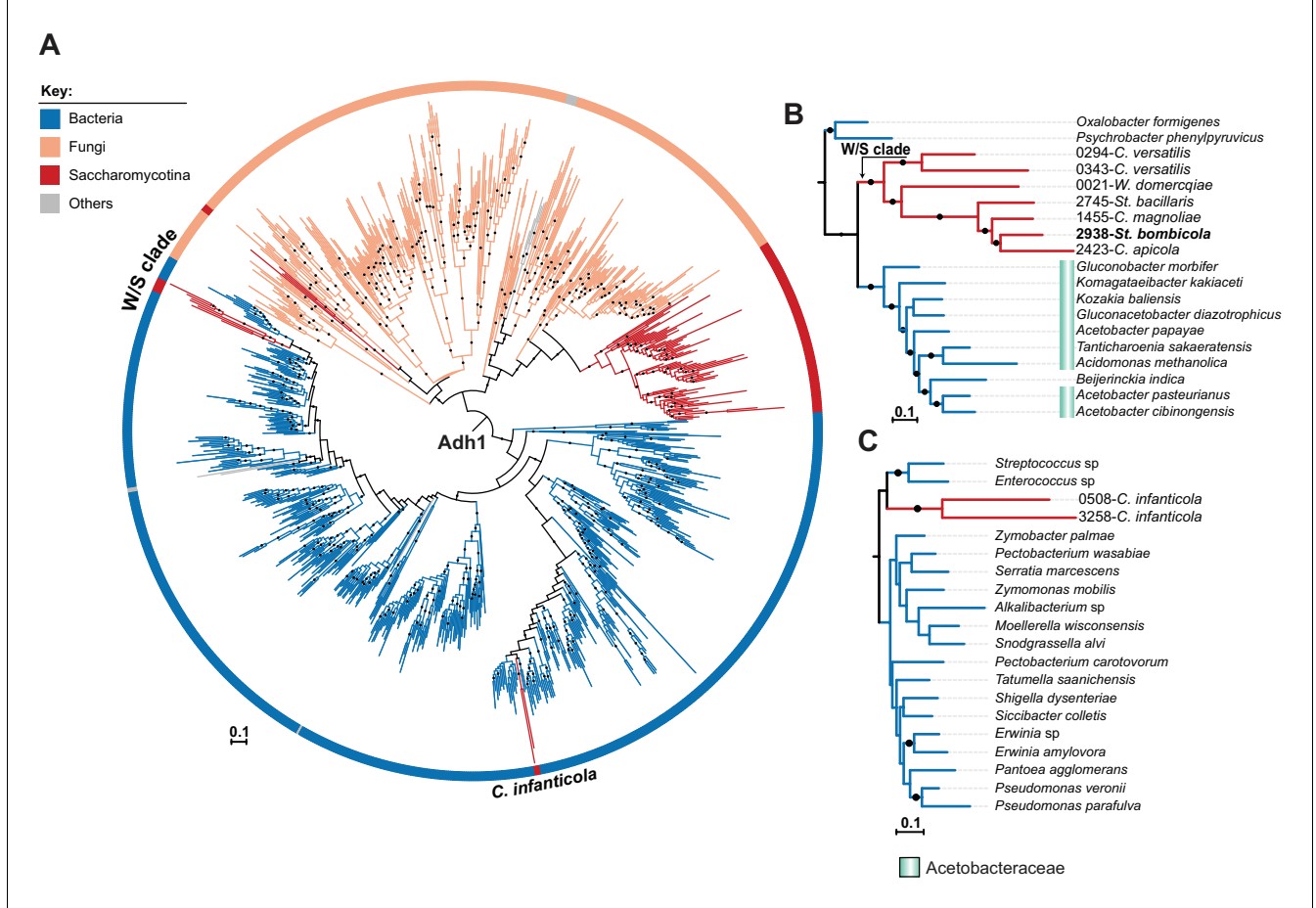

**Figure 5.** ML phylogeny of fungal and bacterial Adh1 proteins. (A) ML phylogeny of Adh1 proteins (top 4000 phmmer hits). W/S-clade species and *C. infanticola* are highlighted. The different lineages are represented with different branch colors (red for Saccharomycotina, blue for bacteria, and orange for other Fungi (i.e. non-Saccharomycotina)). Branches with bootstrap support higher than 90% are indicated by black dots. Poorly represented lineages (<10 sequences) are shown in grey. (B, C) Pruned ML phylogenies of Adh1 depicting the phylogenetic relationship between the W/S clade and Acetobacteraceae (B) and between *C. infanticola* and other groups of bacteria (C). For W/S-clade sequences, the protein ID is indicated before the abbreviated species name.

DOI: https://doi.org/10.7554/eLife.33034.014

The following figure supplements are available for figure 5:

**Figure supplement 1.** Topology test analyses for Adh1 and Suc2.
DOI: https://doi.org/10.7554/eLife.33034.015

**Figure supplement 2.** Alcohol dehydrogenase (Adh) activities.
DOI: https://doi.org/10.7554/eLife.33034.016

**Figure supplement 3.** Growth of wt and *adh1Δ* under aeration and microaeration.
DOI: https://doi.org/10.7554/eLife.33034.017

in bacteria, we set out to expound putative functional differences between Adh xenologs found in the W/S clade and 'native' Adh enzymes. Specifically, we compared alcohol dehydrogenase (Adh) activity in three W/S-clade species, in the closely related yeast *B. adeninivorans*, and in the model species *S. cerevisiae*, as well as in a distantly related fructophilic yeast species *Zygosaccharomyces kombuchaensis*. All W/S-clade species tested were capable of using ethanol as carbon and energy source and have a Crabtree-negative behavior when growing on sugars, meaning that ethanol production in aerated batch cultures starts only when cell densities are very high, limiting oxygen availability (typically OD640 nm 30,~5–10 g/L ethanol in *St. bombicola*).

In all non-W/S-clade species tested, the characteristic NADH-dependent Adh activity was readily observed, but no NADPH-dependent Adh activity was detected (*Figure 5—figure supplement 2A*),

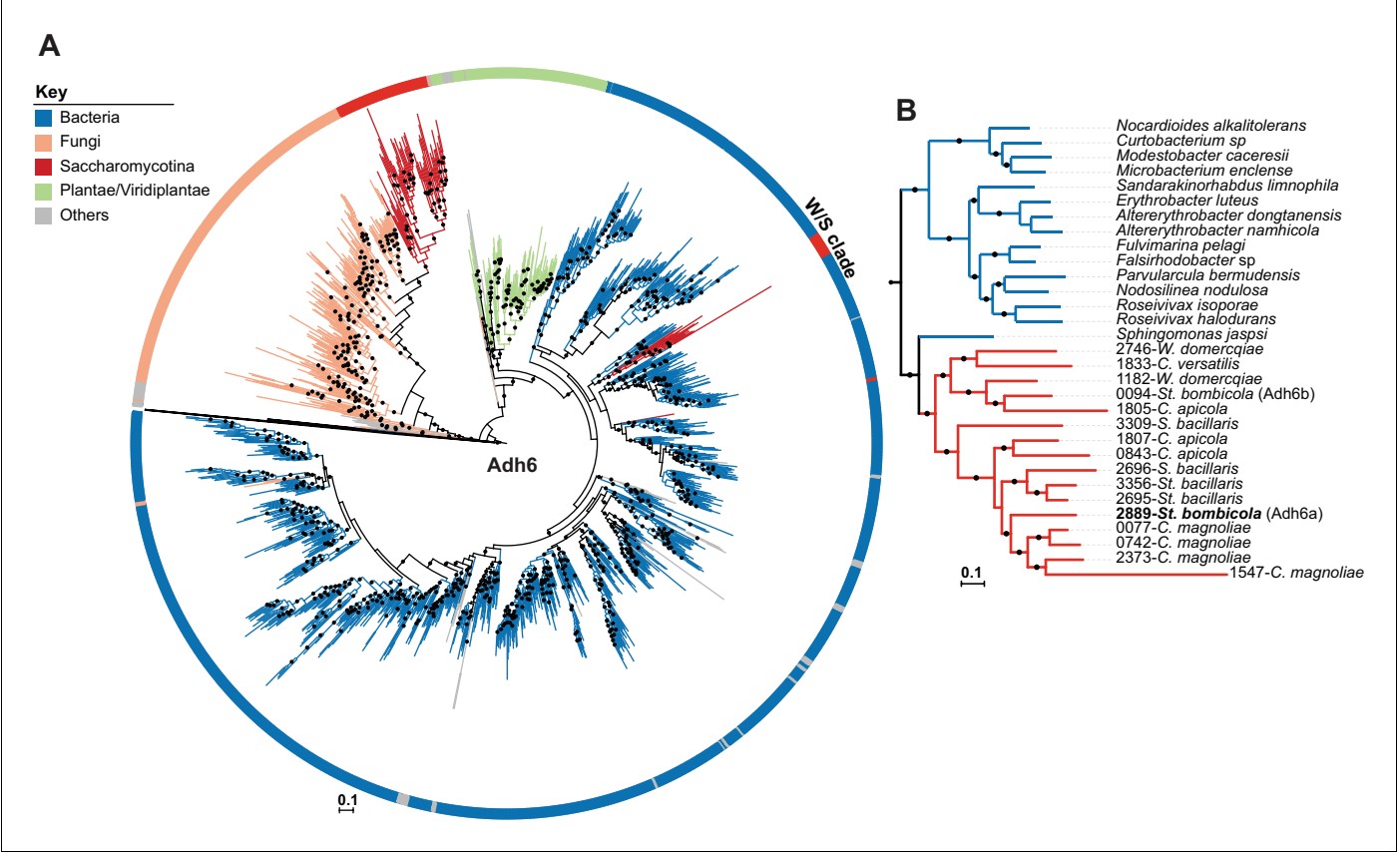

**Figure 6.** ML Phylogeny of Adh6 proteins. (**A**) The phylogeny was constructed with the top 10,000 phmmer hits using *St. bombicola* Adh6 as a query (in bold, Panel B). Sequences with more than 80% similarity were eliminated. Branches with bootstrap support higher than 90% are indicated by black dots. Poorly represented lineages (<10 sequences) are shown in grey. The different lineages are represented with different branch colors (red for Saccharomycotina, blue for bacteria, and orange for other Fungi (i.e. non-Saccharomycotina)). Adh1-like sequences were collapsed as indicated. (**B**) Pruned ML phylogeny depicting the phylogenetic relationship between Adh6 sequences from the W/S clade and their closest bacterial relatives. For W/S-clade sequences, the protein ID is indicated before the abbreviated species name.

DOI: https://doi.org/10.7554/eLife.33034.018

in line with available information concerning yeast enzymes (*Cho and Jeffries, 1998*; *Dashko et al., 2014*; *Ganzhorn et al., 1987*; *Leskovac et al., 2002*). Conversely, all W/S-clade species tested (*St. bombicola*, C. *magnoliae*, and *St. bacillaris*) exhibited Adh activity when either NADH or NADPH was added to the reaction mixture (*Figure 5—figure supplement 2B*). In fact, although both cofactors could be used for conversion of acetaldehyde into ethanol, there was a lower affinity for the substrate (higher $K_m$) for NADPH-dependent activity in *St. bombicola* (*Figure 5—figure supplement 2C*). Interestingly, in *Acetobacter pasteurianus* a bacterial species in the Acetobacteraceae, the same family as the likely donor of W/S-clade Adh1, alcohol dehydrogenase activity was found to be NADH-dependent, although it is unclear whether NADPH was also tested as a cofactor (*Masud et al., 2011*).

## Genetic dissection of alcoholic fermentation in *Starmerella bombicola*

In *S. cerevisiae,* the paralogous enzymes Adh1, Adh2, Adh3, and Adh5 were all shown to contribute to different degrees to the inter-conversion of ethanol and acetaldehyde in a NADH-dependent manner. Although their participation in alcoholic fermentation is not substantial, as may be inferred from the lack of detectable (NADP$^+$ dependent) activity in *S. cerevisiae* crude cell extracts (*Figure 5—figure supplement 2A*), Adh6 and Adh7 can, in principle, also catalyze this type of reaction, using NADPH instead of NADH (*de Smidt et al., 2008*). Since both *ADH1*- and *ADH6*-like genes are present in the genomes of all W/S-clade species studied, it was not clear which enzyme (Adh1-type

or Adh6-type) was responsible for the NADPH-dependent inter-conversion of ethanol and acetaldehyde observed in W/S-clade species. To elucidate this, and to evaluate the impact of alcohol dehydrogenases on metabolism, three deletion mutants were constructed in *St. bombicola* (*adh1Δ*, *adh6aΔ*, and *adh6bΔ*). During aerated growth, deletion of *ADH1* (*adh1Δ*) did not seem to significantly affect specific growth rates in glucose or fructose when compared to the wild type (*Figure 5—figure supplement 3A*). However, we noted a five-fold decrease in ethanol production (*Figure 7A*) and the absence of growth on ethanol as sole carbon and energy source in the *adh1Δ* mutant (*Figure 7—figure supplement 1*). Although some ethanol was produced, no Adh activity was detected in cell-free extracts of the *adh1Δ* mutant when either NADH or NADPH was used (*Figure 5—figure supplement 2E*). These results suggest that Adh1 is the main enzyme used in alcoholic fermentation in *St. bombicola* and that it therefore likely accepts both NADH and NADPH as cofactors. We predicted that, if W/S-clade Adh1 enzymes were mainly used in the recycling of NADPH, which does not normally occur in yeasts, in its absence, compensation would be expected to occur in other $NADP^+$ regenerating reactions. If on the contrary, W/S-clade Adh1 enzymes were mainly used in the recycling of NADH, the compensatory increase of a $NAD^+$ producing reaction would be observed. In line with the cofactor preference measured in cell extracts, the significant decrease in ethanol yield seems to be counterbalanced by a concomitant increase in glycerol production (*Figure 7A*), similarly to what has been observed in the *S. cerevisiae adh1Δ* mutant (*de Smidt et al., 2012*). Moreover, growth of the *adh1Δ* mutant cultivated on identical growth medium but under limited aeration was severely affected (*Figure 5—figure supplement 3B*). Taken together, these observations strongly suggest that Adh1 plays an important role in redox homeostasis, namely in $NAD^+$ regeneration in the absence of oxygen because, similarly to *S. cerevisiae*, glycerol formation in *St. bombicola* is probably a $NAD^+$ regenerating reaction. Consistent with this hypothesis, we did not detect $NADP^+$ dependent glycerol dehydrogenase activity in cell-free extracts (*Figure 5—figure supplement 2D*), and in at least one W/S-clade species, this reaction was shown to be NADH-dependent (*Van Bogaert et al., 2008*). In contrast, mannitol production, which is a $NADP^+$ regenerating reaction (*Lee et al., 2003b*), was significantly decreased in the *adh1Δ* mutant (*Figure 7A*).

The deletion of each of the two *ADH6* paralogous genes (*adh6aΔ* and *adh6bΔ* mutants) did not significantly affect ethanol production (*Figure 7A*). This means that, although some of the enzymes encoded by these genes might be involved in the production of ethanol in the absence of Adh1, when Adh1 is functional, they are not essential for ethanol metabolism and are probably mainly involved in other metabolic reactions. Finally, to ascertain how perturbations in alcoholic fermentation might affect the relative preference of *St. bombicola* for fructose over glucose, we monitored the consumption of both sugars in aerated cultures over time in the *adh1Δ*, *adh6aΔ*, and *adh6bΔ* mutants. There was a significant decrease in sugar consumption rates in the *adh1Δ* mutant (*Figure 7B*), but fructophily was still observed in all mutants and became even more pronounced as the lack of Adh1 affected glucose consumption more than it did fructose consumption.

Orthologs of *S. cerevisiae* Pdc1, which catalyzes the first step in the fermentative pathway, appeared to be absent in W/S-clade genomes. Since all W/S-clade species investigated produce ethanol, which requires prior decarboxylation of pyruvate to acetaldehyde, it follows that the role normally fulfilled by Pdc1 in *S. cerevisiae* must have been taken over by a different enzyme in W/S-clade yeasts. According to our phylogenetic analysis, the only candidate likely to assume this function would be the product of the *ARO10* gene (*Figure 3A*). However, and although it displays some amino acid sequence similarity with Pdc1, *S. cerevisiae* Aro10 displayed extremely low affinity for pyruvate as a substrate (*Kneen et al., 2011*). Therefore, to ascertain whether Aro10 is fulfilling a role in alcoholic fermentation in W/S-clade yeasts, an *ARO10* deletion mutant (*aro10Δ*) was constructed in *St. bombicola*. This mutant failed to produce ethanol, as might be expected in the complete absence of pyruvate decarboxylase activity (*Figure 7A*). Similarly to the *adh1Δ* mutant, it exhibited increased glycerol production, but unlike the former, it grew considerably slower than the wild type strain in aerated conditions (*Figure 7C*). This decrease was probably mainly due to the fact that the absence of pyruvate decarboxylase activity affects other metabolic routes in addition to alcoholic fermentation, such as the production of acetate and acetoin (*Flikweert et al., 1996*). The phenotype observed in this mutant confirmed that Aro10 is the only decarboxylase involved in alcoholic fermentation in *St. bombicola* and that, therefore, the modification of enzymatic specificities was also involved in remodeling alcoholic fermentation in W/S-clade yeasts. Importantly, in the *aro10Δ* mutant, glucose consumption seems to be even more seriously affected than in the *adh1Δ* mutant

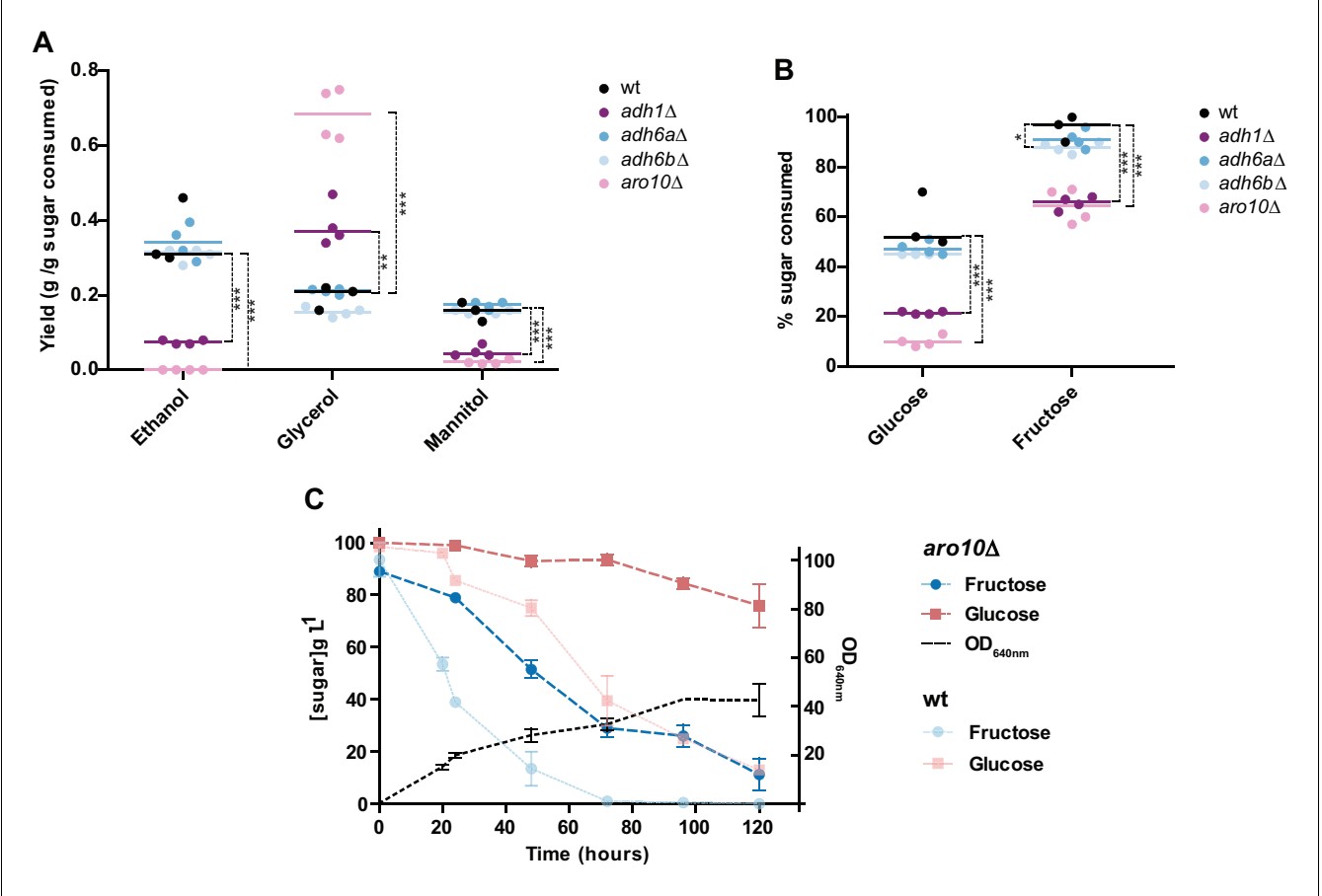

**Figure 7.** Metabolite production and sugar consumption in *St. bombicola* wild type (wt) and deletion mutants (*adh1Δ, adh6aΔ, adh6bΔ,* and *aro10Δ*). (A) Ethanol, glycerol, and mannitol yields determined after 72 hr of growth. (B) Percentage of sugar (fructose and glucose) consumed after 72 hr of growth. Mean values are represented by the colored horizontal lines. Assays were performed in duplicate in two biological replicates. All strains were grown in 20FG medium at 30°C with aeration. Statistically significant differences (one way ANOVA, using the Bonferroni's correction) between wt and deletion mutants for sugar consumption and metabolite production are shown (* p-value<0.05; ** p-value<0.01; *** p-value<0.001). Additional p-values for other pairwise comparisons are shown in *Figure 7—source data 1*. (C) Sugar consumption profile of *aro10Δ* mutant grown in 20FG medium. Sugar consumption profile is also shown for the wt strain as indicated in the key and was previously reported in *Figure 1*. Error bars represent standard deviation of assays performed in duplicate in two biological replicates.

DOI: https://doi.org/10.7554/eLife.33034.019

The following source data and figure supplement are available for figure 7:

**Source data 1.** p-Values and data used to construct the plots.
DOI: https://doi.org/10.7554/eLife.33034.021
**Source data 2.** Primers and strategies used to construct the deletion mutants.
DOI: https://doi.org/10.7554/eLife.33034.022
**Figure supplement 1.** Growth of wt and mutants (*adh1Δ* and *aro10Δ*) in ethanol-based medium.
DOI: https://doi.org/10.7554/eLife.33034.020

(*Figure 7B*) and, in fact, glucose was left all but untouched even after 120 hr of growth (*Figure 7C*), while fructose was almost totally consumed.

## Identification of a yeast lineage devoid of an *ADH1* gene

The present study describes a series of evolutionary events affecting the genes involved in alcoholic fermentation in all the species studied so far in the W/S clade, a lineage of more than 100 species that is very distantly related to *S. cerevisiae*. Two alternative hypotheses can be put forward concerning the order of the events underlying the observed remodeling of the fermentative pathway. Both our comparative genomics data and our experimental results suggest that loss of 'native' *ADH1* and

*PDC1* orthologs preceded acquisition of bacterial counterparts, whose extant functions seem to be similar to the roles normally fulfilled by alcoholic fermentation enzymes in yeasts. However, while we found one species currently placed outside the W/S clade (*C. infanticola*) lacking *PDC1*, which is consistent with loss of this gene having preceded acquisition of the bacterial versions of this gene by *C. versatilis*, all species in our analysis possessed either a 'native' or the bacterial version of the *ADH1* gene. In fact, all publicly available genomes across the entire subphylum Saccharomycotina possess at least one *ADH1* ortholog. Remarkably, sequencing of the genome of *Candida galacta* (*Figure 8*) in the context of a distinct project (the Y1000+ Project sequencing the genomes of all known species of Saccharomycotina; http://y1000plus.org)(*Hittinger et al., 2015*) showed that it lacks both *PDC1* and *ADH1* (*Figure 8*, *Figure 2—source data 3*). The phylogenetic position of this species strongly supports the hypothesis that loss of native *ADH1* preceded the (likely independent) acquisition of the bacterial versions of the gene by the *C. infanticola* lineage and by the MRCA of the W/S clade.

## The invertase of bacterial origin is essential for sucrose assimilation in *St. bombicola*

Our results so far are consistent with the hypothesis that the surge in HGT observed in the MRCA of the W/S clade is related to its adaptation to the high-sugar environment in the floral niche, as exemplified by the acquisition of the Ffz1 fructose transporter from filamentous fungi and by the likely reacquisition of alcoholic fermentation involving HGT for alcohol dehydrogenase and modification of enzyme specificity (of Aro10). Acquisition of a bacterial invertase (*sacC*) (*Martin et al., 1987*) by a

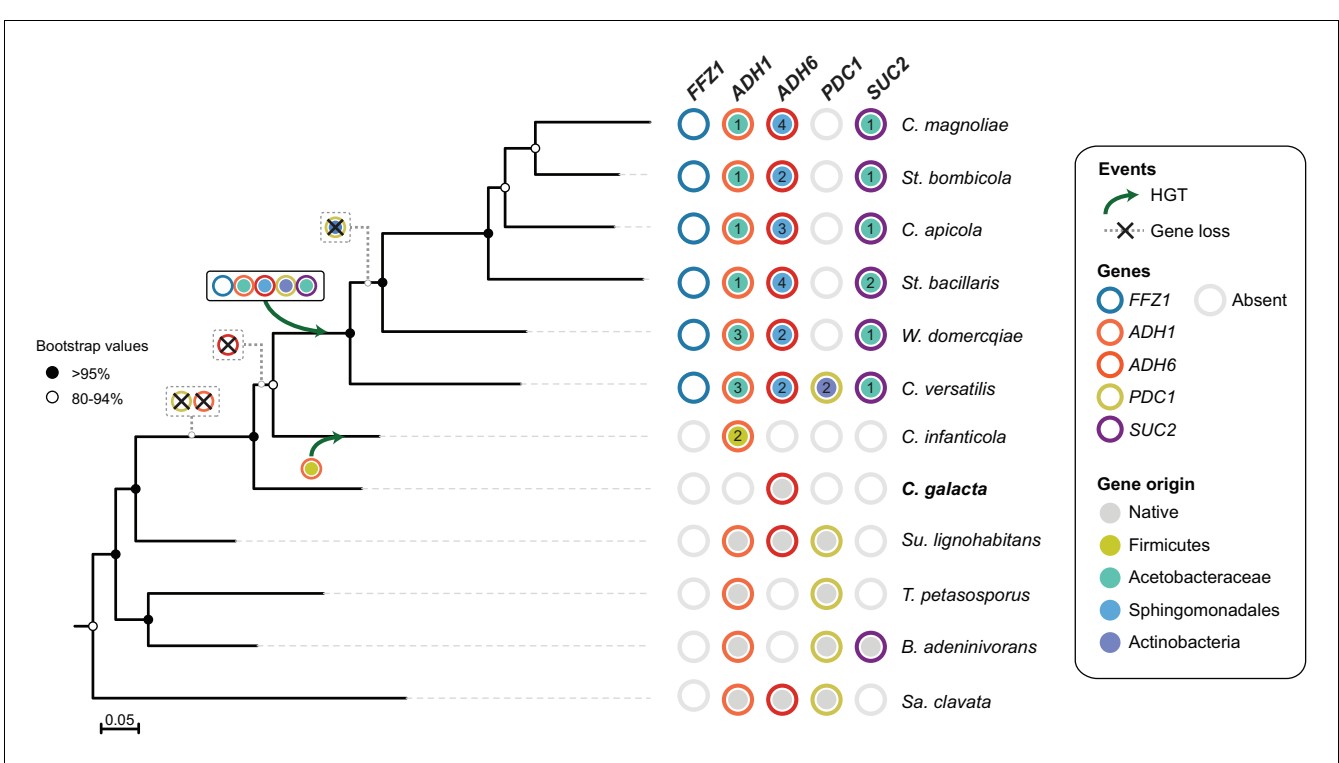

**Figure 8.** Loss and acquisition of sugar metabolism related genes in the W/S clade and closely related lineages. Phylogenetic relationship between W/S species and closest relatives are depicted based on the ML phylogeny using the same dataset as in *Figure 2A*, but with the addition of *C. galacta*. For each of the four relevant genes: *ADH1*, *ADH6*, *SUC2*, and *PDC1*, presence, absence, and the native or bacterial origin of the orthologs found in the cognate draft genomes are shown for each species next to the respective branch of the tree. Each gene is represented by circles with different line colors (blue for previously studied *FFZ1* (*Gonçalves et al., 2016*), orange for *ADH1*, red for *ADH6*, yellow for *PDC1*, and purple for *SUC2*). For xenologs, the different predicted bacterial donor lineages are denoted by different fill colors as indicated in the key. For W/S-clade species, the number of paralogs found in the cognate draft genome is also shown. Inferred gene losses (cross) and HGTs events (arrows) are indicated in the tree using the same color codes.

DOI: https://doi.org/10.7554/eLife.33034.023

lineage lacking this pivotal enzyme for sucrose metabolism is also in line with this hypothesis, since most floral nectars are very rich is sucrose (*Mittelbach et al., 2015*; *Canto et al., 2017*).

To assess whether the horizontally acquired invertase gene is indeed responsible for sucrose assimilation in the W/S clade, a *sacC* deletion mutant (referred henceforth to as *suc2Δ* to emphasize the functional relation to the well-known *S. cerevisiae SUC2* gene) was constructed in *St. bombicola*. Growth assays in medium supplemented with sucrose as sole carbon and energy source, showed that *suc2Δ* mutants were unable to grow (*Figure 4—figure supplement 1A*), while the wild-type strain attained high-cell densities. Furthermore, the *suc2Δ* mutant failed to consume measurable amounts of sucrose, even after 72 hr (*Figure 4—figure supplement 1A*). During growth of the wild-type strain on sucrose, the decrease in sucrose concentrations was accompanied by the appearance of fructose and glucose in the growth medium, strongly suggesting that the horizontally transferred *SUC2* gene encodes an extracellular invertase (*Figure 4—figure supplement 1B*). This conclusion is consistent with the apparent absence of genes encoding sucrose transporters in the W/S-clade genomes analyzed, which indicates that sucrose must be first hydrolyzed outside the cell to be used as a carbon and energy source.

## Discussion

The yeast lineage here named the *Wickerhamiella/Starmerella* (W/S) clade comprises several species that have previously attracted attention due to their unusual metabolic features. The most prominent example is *St. bombicola*, a species used for the production of sophorolipids, which are amphipathic molecules that are employed as biosurfactants (*Samad et al., 2015*; *Takahashi et al., 2011*). *Starmerella bacillaris* is often found in wine fermentations and is known for diverting an important fraction of its carbon flux towards the production of glycerol instead of ethanol (*Englezos et al., 2015*). *Candida magnoliae* has been reported to be capable of producing large amounts of erythritol (*Ryu et al., 2000*). More recently, we reported that fructophily was an important common trait that unified these species and others belonging to the W/S clade, and we also consubstantiated a strict correlation between the presence of the transporter Ffz1 and fructophily. Here, we show that presence of the Ffz1 transporter is a pre-requisite for fructophily in *St. bombicola*, as previously observed in *Z. rouxii* (*Leandro et al., 2014*). The stronger emphasis on the production of sugar alcohols as byproducts of metabolism at the expense of ethanol seemed also to be a common trait between the species examined (*Lee et al., 2003b*; *Ryu et al., 2000*), which led us to hypothesize that the preference of these yeasts for fructose was likely to be part of a broader remodeling of metabolism connected to the adaptation to the high-sugar environments where these yeasts thrive. The present work reflects our effort to uncover other aspects of this adaptation using comparative genomics as a starting point.

Our examination of genes acquired from bacteria showed that the number of HGT events from bacteria into the W/S clade and its neighbor lineage (represented by the species *C. infanticola*), far exceeded the number of events reported for other Saccharomycotina lineages (*Marcet-Houben and Gabaldón, 2010*) and was also considerably higher than those we identified in two species closely related to, but phylogenetically clearly outside the W/S clade (*Su. lignohabitans* and *B. adeninivorans*). The largest number of HGT events was detected in the earliest-diverging species in the W/S clade, *C. versatilis*, which together with the phylogenetic signal in the genes that were acquired through HGT, suggests that a large surge of acquisitions probably occurred in the MRCA of the clade (*Figure 2A*). Under this model, most extant lineages subsequently discarded a large portion of the xenologs originally present in the common ancestor. In line with this hypothesis, it was possible to identify 52 xenolog ortholog groups that were present in at least two W/S-clade species. It seems likely that, in addition to HGT events common only to W/S-clade species, additional HGT events took place in the MRCA of *C. infanticola* and the W/S clade because, from the inspection of the phylogenies constructed for *C. infanticola*, at least six genes of apparent bacterial origin in this species also have bacterial origin in the W/S clade. While the bacterial donor lineage seems to be quite different between *C. infanticola* and the W/S clade for one gene (*ADH1*), strongly suggesting that they were acquired in separate events, the other five genes share a common ancestor, possibly pointing to a single event. As far as can be presently assessed from the output of our AI pipeline, the remaining HGT-derived genes seem to be specific to *C. infanticola*. In addition, we noted that the set of 52 xenologs present in at least two extant W/S-clade species is enriched for genes encoding proteins

that affect redox balance in the cell. In fact, changes in fluxes through main metabolic pathways were previously shown to impact redox balance and oxidative stress in yeasts (*González-Siso et al., 2009*), which is consistent with our hypothesis that associates the acquisition of bacterial genes with adaptive changes in metabolism.

The most striking finding concerning the function of the transferred genes is the profound remodeling of the ubiquitous alcoholic fermentation pathway used by yeasts to convert pyruvate into ethanol with concomitant regeneration of $NAD^+$. The first step of the pathway, consisting of the conversion of pyruvate to acetaldehyde, is normally catalyzed by the enzyme Pdc1. However, in most W/S-clade yeasts, Pdc1 is absent. Here, we provided genetic evidence for one W/S-clade species, *St. bombicola,* that the role of Pdc1 was taken over by a related decarboxylase encoded in *S. cerevisiae* by the *ARO10* gene. In *S. cerevisiae*, phenylpyruvate is the primary substrate of Aro10, which links this enzyme to amino acid catabolism, rather than alcoholic fermentation (*Romagnoli et al., 2012*; *Vuralhan et al., 2005*). However, it has been shown that site-directed mutagenesis of a few selected sites was capable of considerably increasing the affinity of *S. cerevisiae* Aro10 for pyruvate (*Kneen et al., 2011*), which supports the notion that this shift in substrate specificity may have occurred naturally in the course of evolution of W/S-clade yeasts. Intriguingly, *C. versatilis*, also lacks a native *PDC1* gene but possesses two genes of bacterial origin encoding *PDC1* orthologs, which coexist with *ARO10*. In this species, it is possible that the xenologs, and not *ARO10*, carry out the conversion of pyruvate into acetaldehyde.

The second step in alcoholic fermentation is the conversion of acetaldehyde into ethanol, which is conducted in *S. cerevisiae* mainly by Adh1. Again, 'native' *ADH1* genes were absent from all W/S-clade genomes examined, and it seems that the MRCA of the W/S clade acquired a bacterial *ADH1* gene, from which extant *ADH1* xenologs found in all extant W/S-clade species examined were derived. A similar occurrence was detected involving the loss of the 'native' *ADH6* ortholog, encoding a NADPH-dependent branched chain alcohol dehydrogenase and the acquisition of bacterial orthologs, although the bacterial donor lineages of the *ADH1* and *ADH6* xenologs seem to be distinct. We showed that, in *St. bombicola*, the *ADH1* xenolog is absolutely required for growth on ethanol and is also mainly responsible for alcoholic fermentation, thereby performing the functions fulfilled by two different enzymes in *S. cerevisiae* (where Adh2 catalyzes ethanol assimilation). The two *ADH6* xenologs played a minor role, if any, in alcoholic fermentation, similarly to what happens in *S. cerevisiae*.

All our results are consistent with the hypothesis that alcoholic fermentation was first lost in an ancestral lineage and was subsequently reacquired by W/S-clade yeasts through horizontal acquisition of genes. In one instance, a pre-existing yeast gene (*ARO10)* also modified its enzymatic specificities to become involved in alcoholic fermentation. On the other hand, no evidence was found for the alternative hypothesis stating that acquisition preceded loss, such as the co-occurrence in the same genome of 'native' and bacterial Adh1 or a distinctive role for the bacterial enzyme in yeast metabolism, which might have driven the fixation of the bacterial version of Adh1. Loss and subsequent reacquisition of a metabolic pathway through multiple HGT events was reported previously for the unicellular red algae *Galdieria phlegrea* where massive gene loss occurred concomitantly with adaptation to a specialized niche in the common ancestor of Cyanidiophytina red algae (*Qiu et al., 2013*). The lines of evidence supporting a similar event for alcoholic fermentation in the W/S clade are threefold, as follows.

First, one key aspect backing the 'loss followed by reacquisition' hypothesis is the identification of a yeast lineage, here represented by *C. galacta*, which lacks an *ADH1* ortholog, either 'native' or bacterial. The species phylogeny presented in *Figure 8* places this species close to *C. infanticola,* in a position consistent with our prediction for an extant representative of an Adh1- lineage pre-dating the acquisition of bacterial *ADH1* genes. In addition, the two genes forming the alcoholic fermentation pathway seem to have been lost in quick succession because no genome was found that possessed only 'native 'ADH1* or only 'native' *PDC1* genes. This also strongly suggests that loss of the entire pathway pre-dated acquisition of new genes or functions.

While *ADH1* and *ADH6* xenologs persist in all extant W/S-clade species examined, bacterial *PDC1* xenologs, if they were indeed also acquired by the MRCA of the clade, were subsequently lost in most species. These hypothesized losses may have occurred as Aro10 evolved to fulfill the function of Pdc1. *C. versatilis*, which is the earliest-diverging species in the W/S clade, is a notable exception that possesses two *PDC1* xenologs in addition to *ARO10*. Hence, *C. versatilis* on the one

hand, and the remaining W/S clade species on the other hand, seem to represent two distinct solutions restoring pyruvate decarboxylase activity, which is a second observation in line with gene loss being the ancestral event.

Our assessment of the performance of the Adh1 enzymes in W/S-clade species constitutes a third argument in favor of the 'loss followed by reacquisition' hypothesis. We showed that W/S-clade enzymes of bacterial origin possess a potentially advantageous characteristic when compared to their yeast counterparts; they are capable of regenerating both $NAD^+$ and $NADP^+$, while the yeast enzymes accept only NADH as a cofactor. However, at least in *St. bombicola*, elimination of Adh1 was compensated by an increase in glycerol formation, which is very likely a $NAD^+$ regenerating reaction, and not of mannitol formation, which regenerates $NADP^+$. It is therefore reasonable to infer that the Adh xenologs are playing a similar role to that normally fulfilled by the native Adh enzymes. Morever, complete elimination of alcoholic fermentation, as observed in the *aro10Δ* mutant, resulted in an even more pronounced compensation at the level of glycerol production.

In conclusion, given the evidence presented here, the presence of xenologs involved in alcoholic fermentation in extant W/S-clade yeasts can be best explained by the need to restore alcoholic fermentation in a lineage that had previously lost it. In this case, the involvement of bacterial genes seems to have been circumstantial and possibly a consequence of the availability of the donor in the same environment. The ecological opportunity for the horizontal exchange of genetic material underlying the acquisition of bacterial *ADH1* seems to have existed for a long time, since W/S-clade Adh1 proteins are most closely related to those of a bacterial lineage that is also frequently associated with the floral niche (Acetobacteraceae) (*Iino et al., 2012*; *Suzuki et al., 2010*; *Tucker and Fukami, 2014*).

*SUC2* is also among the genes acquired by the MRCA of the W/S clade. Unlike the 'native' version of *ADH1*, which is ubiquitous in the Saccharomycotina except in the W/S clade, evolution of 'native' *SUC2* seems to be punctuated by multiple loss events resulting in a patchy extant distribution (*Carlson et al., 1985*), which is probably linked to its role as a 'social' gene (*Sanchez and Gore, 2013*).

Taken together, our results suggest the enticing hypothesis that the basis for fructophily arose in yeasts in a lineage devoid of alcoholic fermentation, but probably well equipped to use fructose as electron acceptor with concomitant production of mannitol (*Baek et al., 2010*; *Lee et al., 2003a*), through the horizontal acquisition of the Ffz1 transporter that enabled the efficient utilization of fructose as carbon and energy source. Such a lineage would resemble the *aro10Δ* mutant in that it would be unable to use glucose in high-sugar conditions. This putative ancestor might subsequently have acquired fermentative capacity to enable efficient glucose consumption and *SUC2* to permit the utilization of sucrose, thereby completing the set of tools required to use the sugars abundant in the floral niche.

In conclusion, our previous (*Gonçalves et al., 2016*) and present results uncovered, to our knowledge for the first time, an instance of major remodeling of central carbon metabolism in fungi involving the horizontal acquisition of a multitude of genes mostly from bacteria. Thereby, fructophilic yeasts seem to have been able to overcome the inability to use glucose efficiently, as observed in extant fructophilic bacteria. As progress in the Y1000+ Project provides additional genomic information concerning species within and in the vicinity of the W/S clade, the improved phylogenetic resolution will hopefully help further extricate the complex pattern of acquisitions of genes.

## Materials and methods

**Key resources table**

| Reagent type (species) or resource | Designation | Source or reference | Identifiers | Additional information |
| --- | --- | --- | --- | --- |
| Strain, strain background | *Starmerella bombicola* PYCC 5882 | Portuguese Yeast Culture Collection | | |
| Strain, strain background | *Candida magnoliae* PYCC 2903 | Portuguese Yeast Culture Collection | | |

*Continued on next page*

*Continued*

| Reagent type (species) or resource | Designation | Source or reference | Identifiers | Additional information |
|---|---|---|---|---|
| Strain, strain background | *Starmerella bacillaris* PYCC 3044 | Portuguese Yeast Culture Collection | | |
| Strain, strain background | *Saccharomyces cerevisiae* PYCC 7186 (S288C) | Portuguese Yeast Culture Collection | | |
| Genetic reagent (*Starmerella bombicola*) | *adh1Δ* (*adh1Δ::HYG*) | This paper | | Constructed as described in the materials and methods section; primers used are described in *Figure 7—source data 2* |
| Genetic reagent (*Starmerella bombicola*) | *aro10Δ* (*aro10Δ::HYG*) | This paper | | Constructed as described in the materials and methods section; primers used are described in *Figure 7—source data 2* |
| Genetic reagent (*Starmerella bombicola*) | *adh6aΔ* (*adh6aΔ::HYG*) | This paper | | Constructed as described in the materials and methods section; primers used are described in *Figure 7—source data 2* |
| Genetic reagent (*Starmerella bombicola*) | *adh6bΔ* (*adh6bΔ::HYG*) | This paper | | Constructed as described in the materials and methods section; primers used are described in *Figure 7—source data 2* |
| Genetic reagent (*Starmerella bombicola*) | *ffz1Δ* (*ffz1Δ::HYG*) | This paper | | Constructed as described in the materials and methods section; primers used are described in *Figure 7—source data 2* |

## Strains

Yeast strains were obtained from PYCC (Portuguese Yeast Culture Collection, Caparica, Portugal) or NRRL (USDA ARS Culture Collection). All strains were maintained in YPD medium.

## Identification of *ADH1*, *PDC1*, *ADH6*, and *SUC2* genes in the W/S clade and related species

A custom query consisting of *ADH1*, *ADH6*, *PDC1*, and *SUC2* genes from *S. cerevisiae* was used in a tBLASTx search against W/S-clade genomes (*C. versatilis* JCM 5958, *St. bombicola* PYCC 5882, *St. bacillaris* 3044, *C. magnoliae* PYCC 2903, *C. apicola* NRRL Y-50540 and *W. domercqiae* PYCC 3067). The genomes of closely related species *Candida galacta* NRRL Y-17645 (Y1000$^+$ Project), *Sugiyamaella lignohabitans* CBS 10342, *Blastobotrys adeninivorans* LS3, *Trichomonascus petasosporus* NRRL YB-2093, and *Saprochaete clavata* CNRMA 12.647 were also added to the database (*Figure 8*, see *Figure 2—source data 3*). The best tBLASTx hit sequences (E-value $<e^{-10}$) retrieved from the analysis were subsequently blasted against the NCBI non-redundant protein database, and orthology was assumed whenever the best hit in *S. cerevisiae* was the corresponding protein in the query. Genes related with glycolysis were also inspected (*Figure 2—source data 3*). For this analysis, distantly related fructophilic species *Zygosaccharomyces kombuchaensis* (strain CBS 8849) was also added to the database.

For the genes of bacterial origin in the W/S clade, genome location was assessed for each gene in order to discard possible assembly contaminations (as scaffolds containing only bacterial genes). Although assembly artifacts cannot be excluded, all the key genes are located in considerably long scaffolds and flanked by other genes of yeast origin.

Genomes of *C. apicola* (*Vega-Alvarado et al., 2015*) and *C. versatilis* (PRJDB3712) are publicly available, while the draft genomes of the remaining W/S-clade species examined and of *Z. kombuchaensis* were generated in the course of a previous study (*Gonçalves et al., 2016*) and are also publicly available (PRJNA416493 and PRJNA416500). The genome assembly of *Candida galacta* NRRL Y-17645 was obtained in the context of the Y1000+ Project (*Hittinger et al., 2015*) using standardized sequencing (*Hittinger et al., 2010*) and assembly (*Zhou et al., 2016*) protocols and is published here for the first time. This Whole Genome Shotgun project has been deposited at DDBJ/

ENA/GenBank under the accession PPSZ00000000. The version described in this paper is version PPSZ01000000.

## Protein prediction in newly sequenced genomes

For *St. bombicola* PYCC 5882, *St. bacillaris* PYCC 3044, *C. magnoliae* PYCC 2903, *W. domercqiae* PYCC 3067, and *C. infanticola* DS-02 (PRJNA318722) the complete proteome was predicted with AUGUSTUS (*Stanke et al., 2008*) using the complete model and *S. cerevisiae*, *Scheffersomyces stipitis*, and *Y. lipolytica* as references. To assess the completeness of the predicted proteomes in each case, the number of predicted proteins was compared to the number of proteins reported for the annotated proteomes of *W. domercqiae* JCM 9478 (PRJDB3620) and *S. bombicola* JCM 9596 (PRJDB3622). Predictions that used *S. cerevisiae* as a reference turned out to be the most complete (~4.000 predicted proteins) and were therefore used for all downstream analyses. For *C. apicola* NRLL Y-50540, *C. versatilis* JCM 5958, *Su. lignohabitans* (NCBI) and *B. adeninivorans* (JGI), publicly available proteomes were used.

## High-throughput detection of HGT-derived genes from bacteria in the W/S clade

Query protein sequences were searched against a local copy of the NCBI refseq protein database (downloaded May 5, 2017) using phmmer, a member of the HMMER3 software suite (*Eddy, 2009*) using acceleration parameters –F1 1e-5 –F2 1e-7 –F3 1e-10. A custom perl script sorted the phmmer results based on the normalized bitscore (*nbs*), where *nbs* was calculated as the bitscore of the single best-scoring domain in the hit sequence divided by the best bitscore possible for the query sequence (i.e. the bitscore of the query aligned to itself). The top ≤10,000 hits were retained for further analysis, saving no more than five sequences per unique NCBI Taxonomy ID.

The alien index score (AI) was calculated for each query protein (modified from *Gladyshev et al., 2008*). Two taxonomic lineages were first specified: the RECIPIENT into which possible HGT events may have occurred (Saccharomycetales, NCBI Taxonomy ID 4892), and a larger ancestral GROUP of related taxa (Fungi, NCBI Taxonomy ID 4751). The AI is given by the formula: $\mathrm{AI} = (nbsO - nbsG)$, where *nbsO* is the normalized bitscore of the best hit to a species outside of the GROUP lineage, *nbsG* is the normalized bitscore of the best hit to a species within the GROUP lineage (skipping all hits to the RECIPIENT lineage). AI is greater than zero if the gene has a better hit to a species outside of the group lineage and can be suggestive of either HGT or contamination. Note that the original *Gladyshev et al. (2008)* AI calculation was based on relative *E*-values and ranged from −460 to +460, with AI >45 considered a strong HGT candidate. By converting the AI to a bitscore-based metric, the results are not impacted by BLAST version, database size, or computer hardware. The bitscore-based AI score ranges from −1 to +1, with AI >0.1 considered strong HGT candidates.

Full-length proteins corresponding to the top 200 hits (*E*-value <1 × $10^{-10}$) to each query sequence were extracted from the local database using esl-sfetch (*Eddy, 2009*). Sequences were aligned with MAFFT v7.310 using the E-INS-i strategy and the BLOSUM30 amino acid scoring matrix (*Katoh and Standley, 2013*) and trimmed with trimAL v1.4.rev15 using its gappyout strategy (*Capella-Gutiérrez et al., 2009*). Proteins with trimmed alignments < 150 amino acids in length were excluded. The topologies of the remaining proteins were inferred using maximum likelihood as implemented in IQ-TREE v1.5.4 (*Nguyen et al., 2015*) using an empirically determined substitution model and rapid bootstrapping (1000 replications). The phylogenies were midpoint rooted and branches with local support less than 95 were collapsed using the ape and phangorn R packages (*Paradis et al., 2004*; *Schliep, 2011*). Phylogenies were visualized using ITOL version 3.0 (*Letunic and Bork, 2016*).

For the data included in *Figure 2A*, the number of genes for which a bacterial origin was confirmed after inspection of the correspondent phylogenetic tree are shown. AI results per species are shown as in *Figure 2—source data 2*.

We subsequently selected HGT candidates in the W/S clade (*Figure 2—source data 4*) according to two criteria: the putative bacterial ortholog should be present in at least two W/S-clade species, and it should cluster with a bacterial lineage with strong bootstrap support (>90%). Putative HGT-derived genes shared with non-W/S-clade species were excluded. Application of these criteria yielded 200 strong candidate trees, many of which referred to the same ortholog (genes shared by

several W/S species and/or paralogs, as the case of *ADH1* which corresponds to 10 trees). A final number of 52 different ortholog groups was established after collapsing the replicate phylogenies and lineage-specific gene duplications. The resulting set of 52 orthologs was subsequently cross-referenced with GO and InterPro annotations provided by InterproScan and the Joint Genome Institute MycoCosm Portal (*Grigoriev et al., 2014*). KEGG annotation was performed using the KAAS database (*Moriya et al., 2007*). A BLAST KOALA annotation (*Kanehisa et al., 2016*) was also conducted in the final dataset (52 proteins).

## Phylogenetic analyses and topology tests

Species phylogenies were constructed according to *Gonçalves et al. (2016)*. The same dataset was used with the addition of W/S-clade species *C. apicola* NRRL Y-50540; *C. versatilis* JCM 9598; and close relatives *C. infanticola* DS-02, *Candida galacta* NRRL Y-17645 (Y1000$^+$ Project), *Sugiyamaella lignohabitans* CBS 10342 (*Su. lignohabitans*), *Trichomonascus petasosporus* NRRL YB-2093 (*T. petasosporus*), and *Saprochaete clavata* CNRMA 12.647 (*Sa. clavata*). Rpa1, Rpa2, Rpb1, Rpb2, Rpc1 and Rpc2 protein sequences for each species were used to construct the ML tree with RAxML (*Stamatakis, 2006*) v7.2.8 using the PROTGAMMAILG model of amino acid substitution and 1000 rapid bootstraps. Species names abbreviations and accession numbers of the proteins used to construct the phylogeny are indicated in *Figure 2—source data 1*. Phylogenetic relationships are in agreement with the recently published phylogeny of 86 yeast species based on genome sequences (*Shen et al., 2016*).

For the construction of the Pdc1 phylogeny, the top 500 NCBI BLASTp hits from searches against the non-redundant database using Pdc1 from *S. cerevisiae* (CAA97573.1), *St. bombicola* Pdc1-like, and *C. versatilis* Pdc1-like from apparent bacterial origin as queries, were selected. Sequences with more than 90% similarity were removed using CD-HIT v 4.6.7 (*Li and Godzik, 2006*). Pdc1 sequences from the closest relative species *T. petasosporus*, *B. adeninivorans*, and *Sa. clavata* of the W/S clade were also added to the alignment. A total of 479 proteins were aligned using MAFFT v 7.2.15, (*Katoh and Standley, 2014*) using the fast but progressive method (FFT-NS-2). Poorly aligned regions were removed with trimAl (*Capella-Gutiérrez et al., 2009*) using the 'gappyout' option. The ML phylogeny in *Figure 3* was constructed with IQ-TREE v 1.4.3 (*Nguyen et al., 2015*) using the LG + I + G substitution model. For the Suc2 phylogeny (*Figure 4*), the top 200 hits from the phmmer search against the local database were selected and the ML phylogeny was constructed as above. Given the absence of other Saccharomycotina sequences in the top 200 hits for Adh1, the top 4000 hit sequences were selected instead. For Adh6, no Saccharomycotina sequences were found even in the top 4000 phmmer hits, so the top 10,000 top hit sequences were used in this case. For both phylogenies, CD-HIT (*Li and Godzik, 2006*) was used to remove sequences with more than 85% (Adh1) and 80% (Adh6) similarity. For Adh1, a preliminary ML tree was constructed to eliminate sequences outside the Adh1 family. A final set of 976 Adh1 sequences were subsequently aligned and trimmed as aforementioned and used to construct the final Adh1 phylogeny. ML phylogenies were constructed as previously described. Original raw phylogeny files can be accessed using the following links: https://figshare.com/s/c59f135885f31565a864. The likelihood of HGT for *ADH1* and *SUC2* was investigated assuming monophyly of Saccharomycotina as the constrained topology (*Figure 5—figure supplement 1A* and *Figure 5—figure supplement 1C*). The best tree was inferred in RAxML, and ML values for constrained and unconstrained trees were also calculated. The AU test (*Shimodaira, 2002*) implemented in CONSEL (*Shimodaira and Hasegawa, 2001*) was used to compare the unconstrained best tree and the best tree given a constrained topology. To test the independence of Adh1 acquisition, monophily of W/S clade and *C. infanticola* was assumed (*Figure 5—figure supplement 1B*). Phylogenies were visualized using iTOL version 3.0 (*Letunic and Bork, 2016*).

## Detection of enzymatic activities in cell-free extracts

Cultures were grown overnight in YPD medium until late exponential phase (OD$_{640nm}$ ~15–25). Cells were then collected by centrifugation (3000 x g for 5 min), washed twice with cold Tris-HCL buffer (pH = 7.6), and disrupted with glass beads in 500 µL of Lysis Buffer (0.1 M triethanolamine hydrochloride, 2 mM MgCl$_2$, 1 mM DTT and 1 µM PMSF) with six cycles of 60 s vortex-ice. Cell debris were removed by centrifugation at 4°C and 16,000 x g for 20 min and the extracts were stored at −20°C. Alcohol dehydrogenase activity (Adh) assays were performed at 25°C in 500 µL reaction

mixtures containing 50 mM Potassium Phosphate buffer (pH = 7.5), 1 mM of NADH or NADPH, and 25 µL of cell-free extract. The reaction was started by adding acetaldehyde to a final concentration of 100 mM, and reduction of NADH and NADPH was monitored spectrophotometrically by the decrease in absorbance at 340 nm for two minutes. For *St. bombicola* PYCC 5882, 5 mM, 12.5 mM, 50 mM, and 100 mM as final concentrations of acetaldehyde were also used. The absence of Adh activity in the *adh1Δ* mutant was also confirmed with protein extract of *adh1Δ* cells grown in 20FG medium (conditions where ethanol was detected by HPLC in the mutant, *Figure 7A*), using up to three times more protein extract.

For the detection of NADP$^+$-dependent glycerol dehydrogenase activity (*Klein et al., 2017*) in *St. bombicola*, cultures and cell-free extracts were obtained as above. Glycerol dehydrogenase activity was measured in a reaction mixture containing 50 mM Tris-HCl (pH = 8.5) buffer, 1 mm NADP$^+$ and 25 µL of cell-free extract. The reaction was started by adding glycerol to a final concentration of 100 mM, and NADPH formation was monitored spectrophotometrically for two minutes.

## Construction of *St. bombicola* deletion mutants

Standard molecular biology techniques were performed essentially as described in *Sambrook and Russell (2001)* using *E.coli* DH5α as host. *St. bombicola* PYCC 5882 was used in all procedures involving this species. Disruption constructs were designed essentially as outlined by *Van Bogaert et al. (2008)*(*Van Bogaert et al., 2008*). The *St. bombicola GPD* promoter was first amplified by PCR and fused to the hygromycin B phosphotransferase gene (*hygB*) from *E.coli* and the *CYC1* terminator from *S. cerevisiae*. Phusion High Fidelity (Thermo Fisher Scientific, Waltham, MA) was used for *hygB* and *GPD* promoter amplifications. A 491 bp fragment of the GPD promoter (*Van Bogaert et al., 2008*) was amplified using the primer pair GPD_SacI_Fw/GPD_Hind III_Rv (*Figure 7—source data 2*). The *TEF1* promoter from p414TEF-CYC (*Mumberg et al., 1995*) vector was replaced by the *GPD* promoter using *Sac* I and *Hind* III. The primer pair Hyg_Hind III_Fw/Hyg_Xho I_Rv was used to amplify the *hygB* gene from a commercial plasmid (pBlueScript-hyg, [*Niklitschek et al., 2008*]). The amplicon was then cloned into the previously obtained p414GPD-CYC plasmid. The resulting plasmid harbors the hygromycin resistance gene controlled by the *St. bombicola GPD* promoter and followed by the *CYC1* terminator of *S. cerevisiae*.

For disruption of the *ADH1*, *SUC2*, and *ARO10* genes the coding sequences (CDS) with 1 kb upstream and downstream were amplified from genomic DNA using the primer pairs listed in *Figure 7—source data 2*. The two fragments, corresponding to each of the genes, were separately cloned into the PJET1.2 plasmid. The GPD-HYG-CYC cassette was then cloned into each of the resulting PJET1.2 plasmids using the restriction enzymes listed in *Figure 7—source data 2*. For *FFZ1*, *ADH6a*, and *ADH6b*, two sets of primers were used to amplify 1 kb upstream and 1 kb downstream of the CDS. The upstream and downstream fragments of each of the genes were subsequently cloned into the p416GPD-HYG-CYC plasmid using suitable enzymes (*Figure 7—source data 2*), yielding three plasmids each containing a distinct disruption cassette.

*St. bombicola* was transformed by electroporation with each gene disruption construct in turn, amplified by PCR from the respective plasmid template using Phusion High Fidelity DNA polymerase and the primers listed in (*Figure 7—source data 2*), using the protocol described by *Saerens et al. (2011)* (*Saerens et al., 2011*). Two different transformants from each gene disruption transformation were subsequently used for all phenotypic assays.

## Sugar assimilation, consumption, and metabolite production assessments

For metabolite and sugar consumption profiling (*Figure 7A and B*), 10 mL cultures of *St. bombicola* (wild type and mutants) were grown overnight at 30℃ with orbital shaking (180 rpm) in YP medium supplemented with 10% (w/v) of fructose and 10% (w/v) of glucose. The overnight culture was used to inoculate a fresh culture in 30 mL of the same medium to an OD$_{640nm}$ of 0.2, which was incubated under the same conditions. Growth was monitored until late stationary phase was reached (typically after 150 hr), and 2 mL samples were taken at several time points, centrifuged at 12,000 x g for 5 min, and analyzed by HPLC, as previously described (*Gonçalves et al., 2016*). Statistical significance was tested using a one way ANOVA using the Bonferroni's correction for multiple testing, implemented in GraphPad Prism v5. Growth on sucrose and on ethanol was assessed in wild type and

mutants (*suc2Δ*, *adh1Δ* and *aro10Δ*) cultivated overnight in YP medium supplemented with 2% (w/v) sucrose (*suc2Δ*) or 2% (v/v) ethanol (*adh1Δ* and *aro10Δ*) with orbital shaking (180 rpm) at 30°C. Cultures were transferred into the same medium ($OD_{640nm}$ = 0.2), and growth was monitored over time. Consumption of sucrose at different time points was monitored by HPLC.

## Growth assays

*St. bombicola* wild type and *adh1Δ* mutants were tested for growth aerobically (favoring respiration) and in microaerophily (favoring fermentation) conditions. For the mutants, two biological replicates from independent transformations were used. For microaerophily experiments, a 24 hr pre-culture in SC medium supplemented with 0.2% glucose was performed. These cultures were used to inoculate 200 µL of SC medium supplemented with the desired carbon source (2% (w/v) glucose or 2% (w/v) fructose), at a 1:40 ratio in a 96 well plate. The absorbance of each well was read by an unshaken BMG FLUOstar Omega plate reader (*Kuang et al., 2016*) every 120 min at 600 nm for five days. For aerobic growth, a 10 mL pre-culture in SC medium supplemented with 0.2% glucose was performed overnight with shaking (200 rpm). Cells were transferred to 30 mL (in a 250 mL flask) of SC medium supplemented with 2% (w/v) glucose or 2% (w/v) fructose until a final $OD_{640nm}$=0.2. Cultures were grown for 5 days, with shaking.

## Acknowledgements

The authors thank PYCC for providing strains, members of the Yeast Genomics Lab for helpful discussions, Jeremy DeVirgilio and Amanda Beth Hulfachor for technical assistance, the University of Wisconsin Biotechnology Center DNA Sequencing Facility for providing Illumina facilities and services, and Lucigen Corporation (Middleton, WI) for use of their Covaris sonicator. Draft genome data from *C. versatilis* JCM 5958 and *W. domercqiae* JCM 9478 were obtained by RIKEN BioResource Center and RIKEN Center for Life Science Technologies through the Genome Information Upgrading Program of the National Bio-Resource Project of the MEXT, Japan. The authors also thank the US Department of Energy Joint Genome Institute (http://www.jgi.doe.gov) for providing access to genomic data produced in collaboration with the user community. Mention of trade names or commercial products in this publication is solely for the purpose of providing specific information and does not imply recommendation or endorsement by the U.S. Department of Agriculture. USDA is an equal opportunity provider and employer. This work was mainly supported by Unidade de Ciências Biomoleculares Aplicadas-UCIBIO which is financed by national funds from FCT/MEC (UID/Multi/04378/2013) and co-financed by the ERDF under the PT2020 Partnership Agreement (POCI-01–0145-FEDER-007728). This work was also supported by grant PTDC/AGR-ALI/112802/2009 from Fundação para a Ciência e a Tecnologia, Portugal (http://www.fct.pt). CG and MJL are recipients of grants (SFRH/BD/89489/2012 and SFRH/BPD/102803/2014, respectively) from Fundacão para a Ciência e Tecnologia, Portugal. This work was also supported by the National Science Foundation under Grant Nos. IOS-1401682 (JHW), DEB-1253634 (CTH), DEB-1442148 (CTH), and DEB-1442113 (AR); by USDA National Institute of Food and Agriculture Hatch Project 1003258 (CTH); the Robert Draper Technology Innovation Fund from the Wisconsin Alumni Research Foundation (CTH); and funded in part by the DOE Great Lakes Bioenergy Research Center (DOE Office of Science BER DE-FC02-07ER64494 to Timothy J. Donohue). The work was also in part supported through computational resources provided by Information Technology at Purdue, West Lafayette, Indiana; the Center for High Throughput Computing at UW-Madison; and the Advanced Computing Center for Research and Education at Vanderbilt University. CTH is a Pew Scholar in the Biomedical Sciences and a Vilas Faculty Early Career Investigator, supported by the Pew Charitable Trusts and the Vilas Trust Estate. DP is a Marie Sklodowska-Curie fellow of the European Union's Horizon 2020 research and innovation programme, grant agreement No. 747775.

## Additional information

### Funding

| Funder | Grant reference number | Author |
|---|---|---|
| Fundação para a Ciência e a Tecnologia | PTDC/AGR-ALI/112802/2009 | Paula Gonçalves |
| National Science Foundation | IOS-1401682 | Jennifer H Wisecaver |
| National Institute of Food and Agriculture | Hatch Project 1003258 | Chris Todd Hittinger |
| Wisconsin Alumni Research Foundation | | Chris Todd Hittinger |
| DOE Great Lakes Bioenergy Research Center | BER DE-FC02-07ER64494 | Chris Todd Hittinger |
| Pew Charitable Trusts | | Chris Todd Hittinger |
| National Science Foundation | DEB-1253634 | Chris Todd Hittinger |
| National Science Foundation | DEB-1442148 | Chris Todd Hittinger |
| National Science Foundation | DEB-1442113 | Antonis Rokas |
| Fundação para a Ciência e a Tecnologia | UID/Multi/04378/2013 and | Paula Gonçalves |
| European Regional Development Fund | POCI-01-0145-FEDER-007728 | Paula Gonçalves |
| Fundação para a Ciência e a Tecnologia | SFRH/BD/89489/2012 | Carla Gonçalves |
| Fundação para a Ciência e a Tecnologia | SFRH/BPD/102803/2014 | Maria José Leandro |
| H2020 Marie Skłodowska-Curie Actions | 747775 | David Peris |

The funders had no role in study design, data collection and interpretation, or the decision to submit the work for publication.

### Author contributions

Carla Gonçalves, Conceptualization, Data curation, Formal analysis, Investigation, Visualization, Writing—original draft, Writing—review and editing; Jennifer H Wisecaver, Resources, Formal analysis, Investigation, Writing—review and editing; Jacek Kominek, Formal analysis, Investigation, Data acquisition and analysis concerning the Candida galacta genome; Madalena Salema Oom, Supervision, Writing—review and editing; Maria José Leandro, Resources, Supervision, Writing—review and editing; Xing-Xing Shen, Formal analysis, Contributed to topology tests and ontology analysis; Dana A Opulente, Investigation, Data acquisition concerning the Candida galacta genome; Xiaofan Zhou, Formal analysis, Data acquisition concerning the Candida galacta genome; David Peris, Investigation, Writing—review and editing, Data acquisition for Figure 5- figure supplement 3B; Cletus P Kurtzman, Resources, Funding acquisition, Data acquisition concerning the Candida galacta genome; Chris Todd Hittinger, Antonis Rokas, Resources, Supervision, Funding acquisition, Writing—review and editing; Paula Gonçalves, Conceptualization, Resources, Supervision, Funding acquisition, Writing—original draft, Project administration, Writing—review and editing

### Author ORCIDs

Madalena Salema Oom (iD) http://orcid.org/0000-0002-5668-6705
David Peris (iD) http://orcid.org/0000-0001-9912-8802
Chris Todd Hittinger (iD) http://orcid.org/0000-0001-5088-7461
Antonis Rokas (iD) http://orcid.org/0000-0002-7248-6551
Paula Gonçalves (iD) http://orcid.org/0000-0003-2103-1060

Decision letter and Author response
Decision letter https://doi.org/10.7554/eLife.33034.032
Author response https://doi.org/10.7554/eLife.33034.033

## Additional files

### Supplementary files
• Transparent reporting form
DOI: https://doi.org/10.7554/eLife.33034.024

### Major datasets
The following datasets were generated:

| Author(s) | Year | Dataset title | Dataset URL | Database, license, and accessibility information |
|---|---|---|---|---|
| Gonçalves C, Gonçalves P | 2017 | Comparative genomics of fructophilic ascomycetous yeasts | https://www.ncbi.nlm.nih.gov/bioproject/?term=PRJNA416493 | PRJNA416493 |
| Gonçalves C, Gonçalves P | 2017 | Comparative genomics of fructophilic ascomycetous yeasts | https://www.ncbi.nlm.nih.gov/bioproject/PRJNA416500 | PRJNA416500 |
| Kominek J, Opulente DA, Shen XX, Zhou X, Kurtzman CP, Rokas A, Hittinger CT | 2017 | Candida galacta Whole Genome Shotgun project | https://www.ncbi.nlm.nih.gov/bioproject/?term=PRJNA431085 | Publicly available at NCBI BioProject (Accession no. PRJNA431085) |

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
