## [Decision Letter]

Thank you for submitting your article "Evidence for loss and adaptive reacquisition of alcoholic fermentation in an early-derived fructophilic yeast lineage" for consideration by *eLife*. Your article has been favorably evaluated by Patricia Wittkopp (Senior Editor) and three reviewers, one of whom is a member of our Board of Reviewing Editors. The following individual involved in review of your submission has agreed to reveal their identity: Bernard Dujon (Reviewer #3).

The reviewers have discussed the reviews with one another and the Reviewing Editor has drafted this decision to help you prepare a revised submission.

Summary:

The importance of horizontal gene transfer to the evolution of eukaryotes is both underexplored and underappreciated. Here Gonçalves et al. show that components of carbon acquisition and utilization pathways have been replaced, in both sequence and function, by bacterial orthologs in fructophilic yeasts. The results described are an exciting contribution to the field.

Essential revisions:

The reviewers identified two significant matters requiring attention.

1) The title and the Abstract oversell the paper: this over-promising leaves the reader disappointed, rather than appreciative of the work. For example, the Abstract states that "genes required for alcoholic fermentation were lost and subsequently re-acquired from bacteria through horizontal gene transfer" and later indicates that the fermentation pathway was "reinstated", implying that it was lost initially. However, the order of events cannot be established from the results presented. The authors make this clear in the Discussion, writing that "the events leading to the observed profound remodeling of the alcoholic fermentation pathway in W/S-clade yeasts are impossible to trace with certainty because of their antiquity". The evidence required to establish this order – such as a genome without any version ADH1 – is not currently available. This is a shame, since had the authors been able to establish the order of events, the work would be of exceptional interest. Additionally, more evidence would be required to support the conclusion that the observed changes constitute adaptations to a sugar-rich environment as opposed to some other shared feature of the environments occupied by W/S-clade yeasts and not by their close relatives.

2) The narrative. The description of the results is difficult to follow and disorderly (different data sets, different comparisons, and with little guidance to the reader as to where the investigation is going). It would be helpful to lay out from the outset a plausible scenario for the evolution of fructophilic yeast and guide the reader so that s/he engages with the central hypothesis. Next, focus on lateral gene transfer of abilities associated with acquisition of fructose degradation. Report the candidates. Report the evolutionary analyses and the experimental data. Finally, having dealt with gain of new function, it would be appropriate to ask about the ancestral function. Avoid presenting the Results in chronological order: present the strongest results first followed by the secondary findings.

---

## [Author Response]

Essential revisions:The reviewers identified two significant matters requiring attention.1) The title and the Abstract oversell the paper: this over-promising leaves the reader disappointed, rather than appreciative of the work. For example, the Abstract states that "genes required for alcoholic fermentation were lost and subsequently re-acquired from bacteria through horizontal gene transfer" and later indicates that the fermentation pathway was "reinstated", implying that it was lost initially. However, the order of events cannot be established from the results presented. The authors make this clear in the Discussion, writing that "the events leading to the observed profound remodeling of the alcoholic fermentation pathway in W/S-clade yeasts are impossible to trace with certainty because of their antiquity". The evidence required to establish this order – such as a genome without any version ADH1 – is not currently available. This is a shame, since had the authors been able to establish the order of events, the work would be of exceptional interest.

We agree that in the previous version there was a discrepancy between the emphasis given to the “loss before acquisition hypothesis” in the title and Abstract when compared with the Discussion. This pertinent criticism spurred us to review all aspects of the text. It also encouraged us to inspect genomic data concerning species close to the W/S clade generated very recently in the context of the Y1000+ project (https://y1000plus.wei.wisc.edu), and we were fortunate to find a species, *Candida galacta* that lacks any version of *ADH1*. As stated in the review report, this is very important evidence that further strengthens the “loss before acquisition” hypothesis. The finding of this genome was therefore included in the revised version and implied the inclusion of new authors.

In addition, we constructed a new *St. bombicola* mutant in which the *ARO10* gene was deleted. The inability of this mutant to produce ethanol provided experimental confirmation that *PDC1* was lost and that *ARO10* currently fulfills its role in alcoholic fermentation. Since our data also suggests that *ADH1* and *PDC1* were lost in quick succession, because no genome was found possessing either only “native” *ADH1* or only “native” *PDC1* alone, the case for an ancestral loss event is presently soundly supported.

The statement in the previous version quoted in the review report was modified to convey the message that the reconstruction of these kind of ancient events always carries some uncertainty, which was the original intention.

Additionally, more evidence would be required to support the conclusion that the observed changes constitute adaptations to a sugar-rich environment as opposed to some other shared feature of the environments occupied by W/S-clade yeasts and not by their close relatives.

We changed the title and Abstract accordingly. In the remainder of the text we do pinpoint the results consistent with adaptation to a sugar-rich environment, which we still consider to be the most likely driver of the unusual evolutionary events reported. The strong and clear phenotype observed for the *aro10* mutant allowed us to put forward an hypothesis stating how the various events may have contributed to improve sugar metabolism, which is now presented at the end of the Discussion.

2) The narrative. The description of the results is difficult to follow and disorderly (different data sets, different comparisons, and with little guidance to the reader as to where the investigation is going). It would be helpful to lay out from the outset a plausible scenario for the evolution of fructophilic yeast and guide the reader so that s/he engages with the central hypothesis. Next, focus on lateral gene transfer of abilities associated with acquisition of fructose degradation. Report the candidates. Report the evolutionary analyses and the experimental data. Finally, having dealt with gain of new function, it would be appropriate to ask about the ancestral function. Avoid presenting the Results in chronological order: present the strongest results first followed by the secondary findings.

We thoroughly remodeled data presentation, in particular in the beginning of the manuscript. The order in which the results are presented and the titles within the Results section were also changed. In the revised version we used as a thread the comparison with fructophilic bacteria, to which we return throughout the entire manuscript, including a discussion on differences and similarities between the fructophilic bacteria and fructophilic yeasts.